# Evidence for vagal sensory neural involvement in influenza pathogenesis and disease

Nathalie A. J. Verzele[1,2], Brendon Y. Chua[3], Kirsty R. Short[1,4], Aung Aung Kywe Moe[5], Isaac N. Edwards[1], Helle Bielefeldt-Ohmann[1,4], Katina D. Hulme[1], Ellesandra C. Noye[1], Marcus Z. W. Tong[1], Patrick C. Reading[3,6], Matthew W. Trewella[2], Stuart B. Mazzone[2], Alice E. McGovern[2]*

1 School of Chemistry and Molecular Biosciences, The University of Queensland, St Lucia Queensland, Australia, 2 Department of Anatomy and Physiology, The University of Melbourne, Parkville, Victoria, Australia, 3 The Peter Doherty Institute for Infection and Immunity, Department of Microbiology and Immunology, University of Melbourne, Melbourne, Victoria, Australia, 4 Australian Infectious Diseases Research Centre, The University of Queensland, St Lucia, Queensland, Australia, 5 Department of Medical Imaging and Radiation Sciences, Monash University, Clayton, Victoria, Australia, 6 WHO Collaborating Centre for Reference and Research on Influenza, Victorian Infectious Disease Reference Laboratory, Peter Doherty Institute for Infection, and Immunity, 792 Elizabeth St., Melbourne, Victoria, Australia

* alice.mcgovern@unimelb.edu.au

**Data Availability Statement:** Data will be made freely available via Figshare (University of Melbourne online data repository). The location

## Abstract

Influenza A virus (IAV) is a common respiratory pathogen and a global cause of significant and often severe morbidity. Although inflammatory immune responses to IAV infections are well described, little is known about how neuroimmune processes contribute to IAV pathogenesis. In the present study, we employed surgical, genetic, and pharmacological approaches to manipulate pulmonary vagal sensory neuron innervation and activity in the lungs to explore potential crosstalk between pulmonary sensory neurons and immune processes. Intranasal inoculation of mice with H1N1 strains of IAV resulted in stereotypical antiviral lung inflammation and tissue pathology, changes in breathing, loss of body weight and other clinical signs of severe IAV disease. Unilateral cervical vagotomy and genetic ablation of pulmonary vagal sensory neurons had a moderate effect on the pulmonary inflammation induced by IAV infection, but significantly worsened clinical disease presentation. Inhibition of pulmonary vagal sensory neuron activity via inhalation of the charged sodium channel blocker, QX-314, resulted in a moderate decrease in lung pathology, but again this was accompanied by a paradoxical worsening of clinical signs. Notably, vagal sensory ganglia neuroinflammation was induced by IAV infection and this was significantly potentiated by QX-314 administration. This vagal ganglia hyperinflammation was characterized by alterations in IAV-induced host defense gene expression, increased neuropeptide gene and protein expression, and an increase in the number of inflammatory cells present within the ganglia. These data suggest that pulmonary vagal sensory neurons play a role in the regulation of the inflammatory process during IAV infection and suggest that vagal neuroinflammation may be an important contributor to IAV pathogenesis and clinical presentation. Targeting these pathways could offer therapeutic opportunities to treat IAV-induced morbidity and mortality.

can be found via this link https://figshare.com/s/c349583e6a11cdbc0c60.

**Funding:** This study was funded, in part, by grants obtained by A.E.M and S.B.M (2020/GNT2002765) from the National Health and Medical Research Council of Australia, and B.Y.C and A.E.M from the University of Melbourne, Australia. The funders had no role in study design, data collection and analysis, decision to publish, or preparation of the manuscript.

**Competing interests:** The authors have declared that no competing interests exist.

## Author summary

Influenza viruses are a common respiratory pathogen that represent a constant and pervasive threat to human health. Although the inflammatory and immune responses to influenza viral infections are well described, little is known about the role the nervous system plays in the formation and progression of disease. The lungs receive a rich supply of sensory nerve fibers from the vagus nerve. These nerves are critical for protecting the lungs against harmful stimuli and play an important defence role against pathogens, including viruses. Here we use several complex animal models to demonstrate the impact lung sensory neurons have on influenza viral infection and disease outcome. We demonstrate that ablation of lung sensory neurons and inhibition of their neural activity significantly worsens the clinical outcome in mice infected with influenza virus, however with only a moderate impact on lung pathology. Interestingly, when the activity of these neurons is inhibited during influenza viral infection, this drives a hyper neuroinflammatory response within the vagal sensory ganglia, where their cell bodies are located. Our work provides new insights into how these lung sensory neurons are involved in influenza viral infections and may offer therapeutic opportunities to treat influenza-induced morbidity and mortality.

## Introduction

Influenza A virus (IAV) is a common and highly contagious respiratory virus, causing significant morbidity and mortality in humans worldwide [1,2]. Mild IAV infections present with a range of symptoms including a sudden onset of fever, chills, sneeze, cough, congestion, headaches, malaise, and lethargy which generally resolve soon after viral clearance [1,2]. However, in some individuals, including those considered at risk, infections can be life threatening resulting in clinically severe IAV disease associated with pulmonary edema, hypoxemia, pneumonia, and acute respiratory distress syndrome with symptomatology persisting well beyond the clearance of the viral infection.

Although it is well accepted that the pulmonary inflammatory response to IAV is mediated by a diverse range of both resident and recruited cells, much of this *a priori* knowledge fails to recognize the important role played by the nervous system, and in particular sensory neurons, in pulmonary inflammation and disease symptomology. From the larynx to the lung parenchyma, the mammalian respiratory system is innervated by sensory neurons that are almost exclusively derived from the vagus nerve. These vagal sensory neurons are critical for monitoring the pulmonary environment and upon activation transmit information to the brainstem that is then used to drive reflexes and respiratory behaviours [3,4]. These sensory neurons are not homogeneous in phenotype, but rather different subtypes exist, classified by their neurochemistry, molecular expression profiles, physiological properties, and the reflexes that they initiate [4–7].

Some vagal sensory neurons are responsive to potentially damaging stimuli including inhaled, aspirated or locally produced chemicals. These sensory neurons often express receptors for a wide range of immune cell derived molecules [4–6,8] and can be directly activated or modulated by inflammation [4,7–12]. In this regard, vagal sensory neural pathways may be particularly important in the generation of symptoms during respiratory viral infections as they provide input to diverse brain neural circuits that regulate respiratory reflexes, breathing, mood and other complex behaviors [13–18]. We have previously shown that during an acute

infection with IAV, the pulmonary vagal sensory neurons undergo transcriptional changes and take on a neuroinflammatory phenotype. This neuropathy is characterized by an infiltration of immune cells into the vagus nerve and the upregulation of genes in the sensory neurons associated with host defense and inflammation [19]. In addition, these neurons respond by translocating the alarmin high mobility group box-1 (HMGB1) from the nucleus to the cytoplasm, which could serve as a mediator of hyperinnervation and prolonged hypersensitivity by promoting vagal sensory neurite growth and excitability [20].

Interactions between pathogens, inflammation and sensory nerves are unlikely to be unidirectional and recent studies have shown that the nervous system can reciprocally regulate immune processes associated with respiratory disease, notably via pathways involving neuropeptide-containing pulmonary sensory neurons [21–23]. Here, we investigated this interaction further using a combination of surgical, genetic, and pharmacological approaches to assess the effects of interrupting pulmonary sensory neuroimmune interactions on IAV pathogenesis, vagal neuroinflammation and disease.

## Results

### Vagal denervation impacts IAV pathogenesis and disease

We initially sought to characterize the relative contribution of left and right vagus nerves with respect to the sensory innervation to left and right lungs (Fig 1A). Following injection of the retrograde serotype adeno-associated virus encoding tdTomato (AAVrg$^{TdT}$) into the left lung, 75.4% of labelled neurons were located in the left vagal ganglia while 24.6% were in the right vagal ganglia (Fig 1B and 1B'), indicating the left lung receives significantly more sensory innervation from the left compared to the right vagal ganglia. In comparison, the vagal sensory neurons contributing to the right lung appeared to be more evenly distributed across right (59.1%) and left (40.9%) vagal ganglia. Unilateral cervical vagotomies performed 2 weeks prior to injection of AAVrg$^{TdT}$ into either left or right lung eliminated traced neurons in the ipsilateral ganglia only (S1A Fig), indicative that reinnervation does not occur within this time frame. In all experiments included in the analysis, the injection site was contained to the injected lung, with no observable spread of AAVrg$^{TdT}$ into the contralateral lung or pleural cavity.

Considering these findings, we opted to perform both right and left unilateral vagotomies in separate animal cohorts and compared the results to sham vagotomized controls (Fig 1C). Surgical removal of a portion of either the right or left cervical vagus resulted in an initial and transient reduction in body weight compared to sham controls (S1B Fig). Once mice returned to their pre-surgical body weight (~2 weeks) they were intranasally inoculated with 50 plaque forming units (PFU) of the mouse adapted H1N1 IAV, strain A/Puerto Rico/8/34 (PR8). Control (mock infected) mice received intranasal phosphate buffered saline (PBS). No differences in bodyweight were observed in mock infected mice of both right/ left sham and right/ left unilateral vagotomized groups (S2A and S2B Fig). IAV infected mice (both right/ left sham and vagotomized groups) began to lose body weight around days 3 to 4 post-infection, with vagotomized mice losing significantly more weight than their sham IAV infected counterparts (Fig 1D and 1E).

Lung viral titers remained consistent between both right/ left vagotomized and sham vagotomized IAV infected mice at all time points investigated (Figs 2A and 3A). We did not observe any differences between lung cytokines and inflammatory cell infiltrate between mock infected mice of both right/ left sham and right/ left vagotomized groups (S2C and S2D Fig). The induction of cytokines (IL-6, IFNγ, TNFα) observed in lung lysates of right/ left vagotomized IAV infected mice showed some differences compared to sham vagotomized IAV

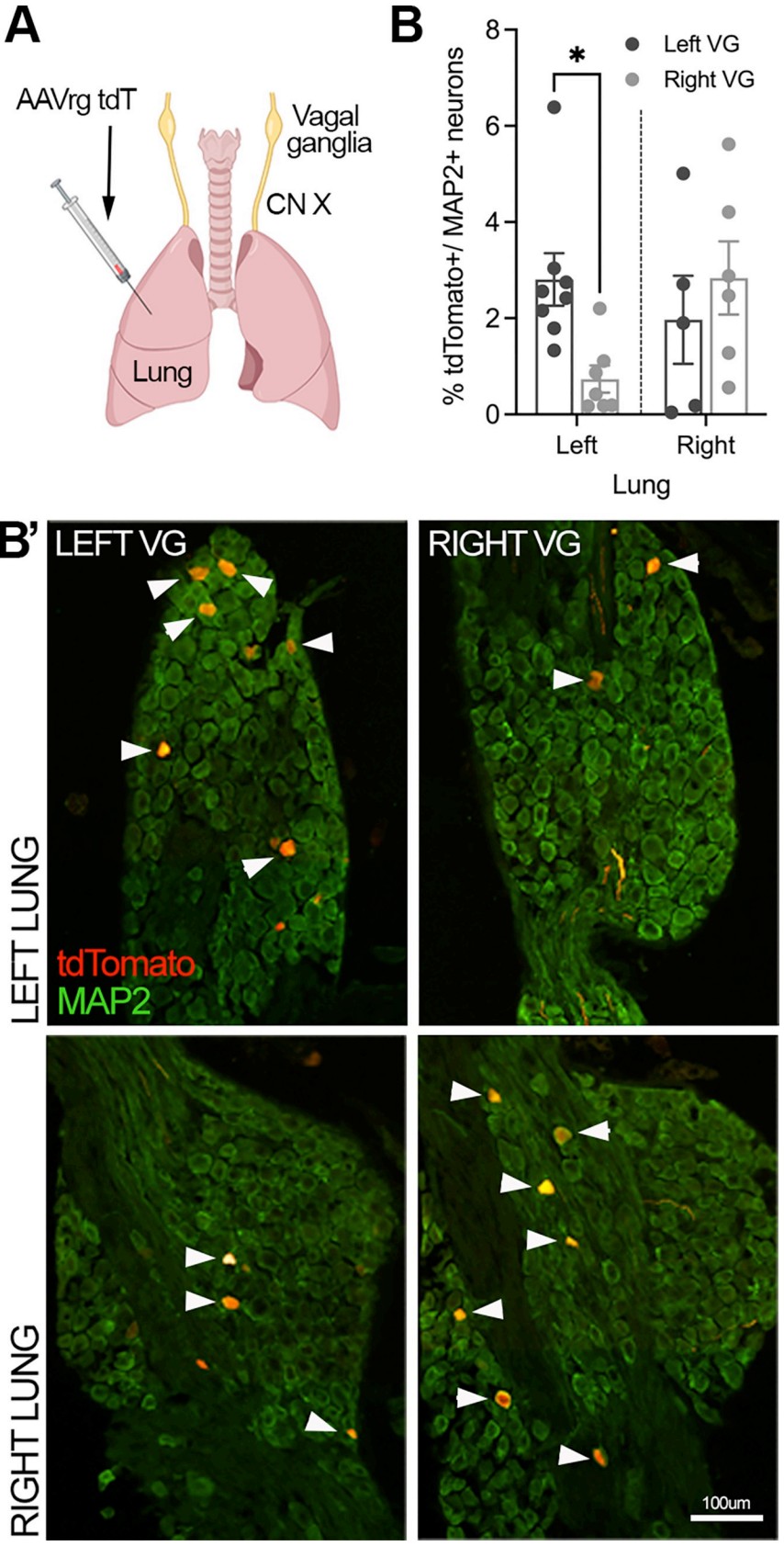

**Fig 1. Characterization of the left and right vagal sensory ganglia innervation to the lungs.** (A) Schematic showing retrograde viral tracing with AAVrg$^{tdT}$ to quantify the ratio of right and left lung projecting sensory neurons in the ipsilateral versus contralateral vagal sensory ganglia. (B) Quantification of the percentage of tdTomato+ neurons per MAP2+ neurons in either left or right vagal sensory ganglia innervating the left (n = 8) or right lung (n = 6) with (B') representative images depicting examples of traced neurons in the vagal sensory ganglia. Data represented as mean ± SEM. * denotes significance of $p < 0.05$, as determined by two-way ANOVA corrected for multiple comparisons (Šídák). CN X, vagus nerve; VG, vagal ganglia. Scale bar represents 100 μm. Cartoons were created with BioRender.com.

infected mice. TNFα and IL-6 were significantly raised in right vagotomized mice at days 3 and 8 post-infection, respectively (Fig 2B), and left vagotomized mice showed an increase in IFNγ at day 8 post-infection (Fig 3B). In addition, we observed occasional significant differences between immune cells in the bronchoalveolar lavage fluid (BALF) of both right/ left vagotomized IAV infected mice compared to their sham groups. In right vagotomized mice, this included an increase in number of natural killer (NK) cells (day 3 post-infection), alveolar macrophages (AM) and B cells (day 8 post-infection), and the reduction of CD4 and NK T-cells at days 3 and 5, respectively (Fig 2C). For left vagotomized IAV infected mice, we observed a significant increase in the number of interstitial macrophages (IM), B, NK and NK T cells at day 8 post-infection (Fig 3C). No difference was observed in systemic cytokine levels between IAV infected vagotomized and sham mice (S3 Fig).

Although the majority of axons comprising the vagus nerve are derived from sensory neurons, a sizeable portion (up to 40%) of the vagal lung innervation originates from parasympathetic motor neurons [24]. In addition, the vagus is not solely a pulmonary nerve but rather innervates numerous other viscera. Therefore, vagotomy is a non-specific method of targeting the sensory innervation supplying the lungs. For more specificity, we developed a genetic approach to selectively ablate lung projecting vagal sensory neurons. In mice, the majority of sensory innervation in the lungs arise from the nodose portion of the vagal sensory ganglia [4,25–27]. The homeodomain transcription factor Phox2B is expressed in nodose neurons and is not present in other neurons that make up the vagal sensory ganglia [28,29]. We utilized a retrograde AAV encoding Cre-recombinase (Cre) dependent diphtheria toxin A (DTA) (AAVrg$^{mCh-FLEX-DTA}$) and initially sought to characterize the construct via unilateral intraganglionic injections into the vagal ganglia of Phox2B Cre-expressing (Phox2B-Cre$^+$) mice (S4A Fig). Four weeks following intraganglionic injection, AAVrg$^{mCh-FLEX-DTA}$ resulted in a significant reduction in the number of neurons expressing neurofilament 200kD and tyrosine hydroxylase neurons, proteins found predominately in nodose neurons [7,25,30,31], within the injected (right side) vagal sensory ganglia in Phox2B-Cre$^+$ mice compared to Phox2B-Cre$^-$ mice (S4B and S4C Fig). Nodose afferents terminate centrally within the medullary nucleus of the solitary tract (NTS) and express the purinergic receptor 2 (P2X2) on their nerve terminals [15,16,32]. Following DTA-mediated nodose neuron ablation we observed a reduction in the density of P2X2-expressing nerve fibers within the ipsilateral NTS in Phox2B-Cre$^+$ mice compared to Phox2B-Cre$^-$ mice (S5 Fig). Vagal preganglionic neurons arising from the brainstem also express Phox2B [33] and we noted a significant reduction in the number of preganglionic neurons within the ipsilateral dorsal motor nucleus of the vagus following intraganglionic injections of AAVrg$^{mCh-FLEX-DTA}$ (dmnx; Phox2B-Cre$^-$: 420.7 ± 49.0 and Phox2B-Cre$^+$: 228.8 ± 31.9 number of choline acetyl transferase (CHAT) positive neurons in the right dmnx, $p = 0.02$). This observation suggests uptake of the AAV construct via preganglionic axons of passage traversing the vagal sensory ganglia, retrograde transport to preganglionic nuclei in the brainstem, and subsequent cre-dependent DTA cellular ablation (S5 Fig). Collectively, these data confirm the retrograde mobility and effectiveness of the viral construct.

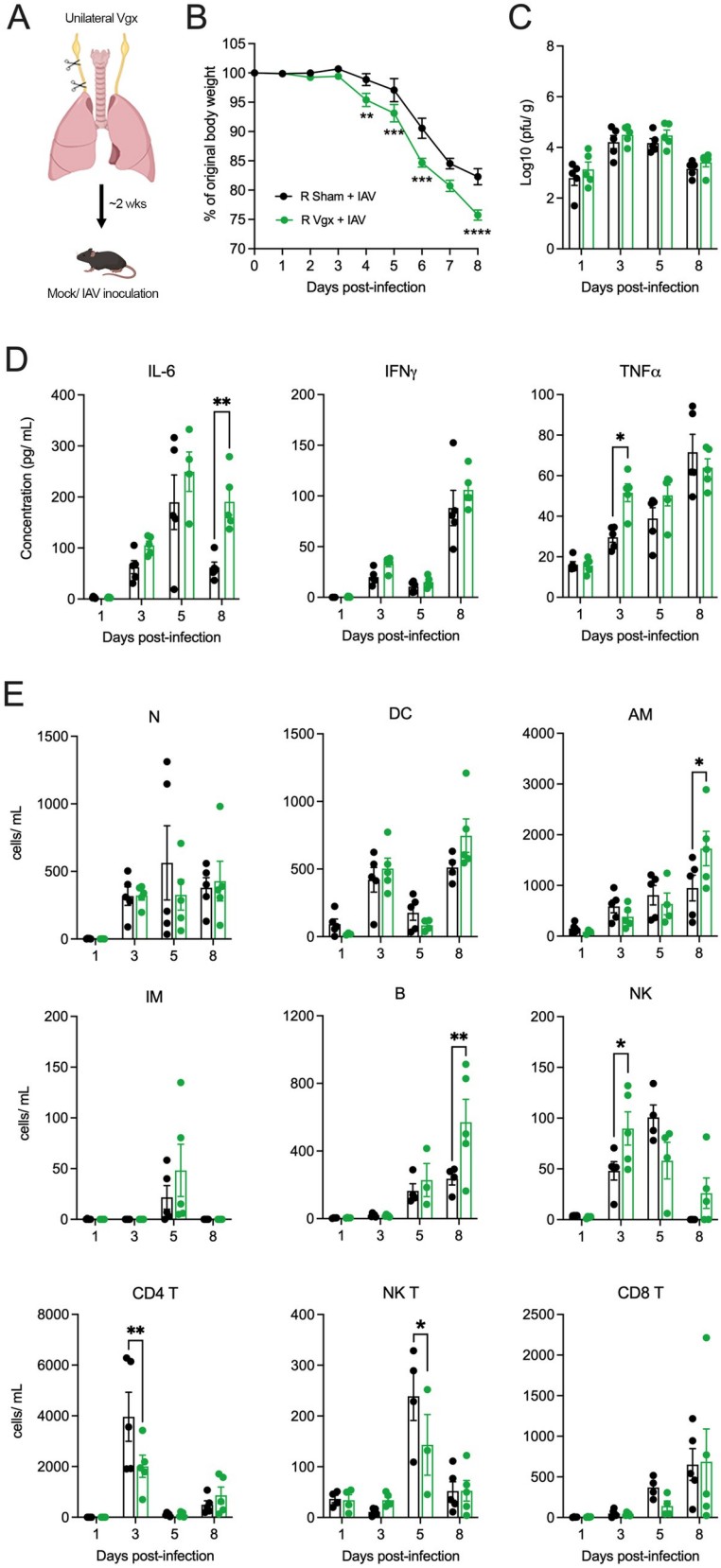

**Fig 2. The impact of right vagotomy on IAV induced pathogenesis.** (A) Schematic outlining the procedure for surgically removing partial vagal innervation to the lungs (right unilateral vagotomy). Graphs depict group level (B) body weight change, (C) lung viral titers, (D) lung cytokine measurements and (E) lung immune cell populations over the course of IAV infection following either right vagotomy (Vgx) or sham surgery (n = 5 per group at days 1, 3, 5, 8 post IAV infection). Data represented as mean ± SEM. *, **, ***, **** denotes significance of $p < 0.05$, $p < 0.01$, $p < 0.001$, $p < 0.0001$, respectively, as determined by repeated measures two-way ANOVA corrected for multiple comparisons (Šídák). N, neutrophils; DC, dendritic cells; AM, alveolar macrophages; IM, interstitial macrophages; B, B cells; NK, natural killer cells. Cartoons were created with BioRender.com.

In subsequent experiments, delivery of AAVrg[mCh-FLEX-DTA] into the lungs of Phox2B-Cre[+] or Phox2B-Cre[-] mice (Fig 4A), was associated with a significantly lower number of mCherry-expressing neurons in Phox2B-Cre[+] mice compared to the Phox2B-Cre[-] littermates (Fig 4B and 4B'). We also observed less mCherry-expressing nerve fibers in the NTS in Phox2B-Cre[+] mice compared to Phox2B-Cre[-] mice (Fig 4B") with no evidence of uptake into preganglionic nuclei populations (dmnx and nucleus ambiguus) indicative of correct AAVrg[mCh-FLEX-DTA] functionality following lung delivery. Similar to surgical vagotomy, cellular ablation of nodose pulmonary afferents did not appear to impact body weight or respiratory function in Phox2B-Cre[+] mice compared to Phox2B-Cre[-] mice prior to infection (S6A and S6B Fig) and no differences in body weight and lung immune cell populations were observed in mock infected Phox2B-Cre[+] and Phox2B-Cre[-] mice (S7A and S7B Fig). Phox2B-Cre[+] mice receiving intrapulmonary injection of AAVrg[mCh-FLEX-DTA] lost significantly more body weight and displayed a more severe clinical score compared to the Phox2B-Cre[-] AAVrg[mCh-FLEX-DTA] group following IAV infection (Fig 4C and 4D). However, like vagotomy, this loss in body weight and severity of clinical symptoms was not associated with differences observed in lung viral titers (Fig 4E), inflammatory cytokines (Fig 4F) or immune cell infiltrates (Fig 4G).

The results of the unilateral vagotomy and AAVrg[mCh-FLEX-DTA] experiments suggest that disruption of the vagal sensory innervation to the lungs impacts the clinical presentation of IAV disease severity. Although modest effects on IAV-induced pulmonary pathology are evident when vagal sensory innervation to the lungs is reduced, it is questionable if these changes in pulmonary pathology are the principal cause of the altered clinical presentation.

## Pharmacological inhibition of the activity of pulmonary sensory neurons during IAV infection

Surgical and DTA-mediated denervation of vagal sensory neurons prior to IAV infection may conceivably cause changes in normal respiratory, immune, and nervous system functions, we shifted to using a pharmacological approach to temporarily inhibit lung afferent activity during IAV infection. We also expanded the range of endpoints measured. QX-314 (N-ethyl-lidocaine) is a charged sodium channel inhibitor that enters subsets of neurons through large-pore ion channels, notably including transient receptor potential vanilloid and ankyrin 1 (TRPV1 and TRPA1). These channels are selectively expressed in populations of vagal sensory neurons involved in airway defense and effectively gated (opened) by products of inflammation. Consequently, QX-314 has been used to inhibit the peripheral induction of action potential formation in sensory neurons in a range of pulmonary inflammatory conditions [21,34,35]. Mice were inoculated with either mock or the H1N1 IAV strain (Auck/09, 5.5 x 10³ PFU) and nebulized with 300μM of QX-314 or vehicle (sterile saline), twice-daily from 3 days post-infection, during the early inflammatory phase when open-probability of sensory nerve fiber TRPV1/ A1 channels is likely increased (Fig 5A) [21,36,37]. Importantly, we observed no difference in body weight and clinical score (S8A and S8B Fig) in mock infected groups receiving either QX-314 or vehicle (PBS). Additionally, baseline respiratory function remained unchanged

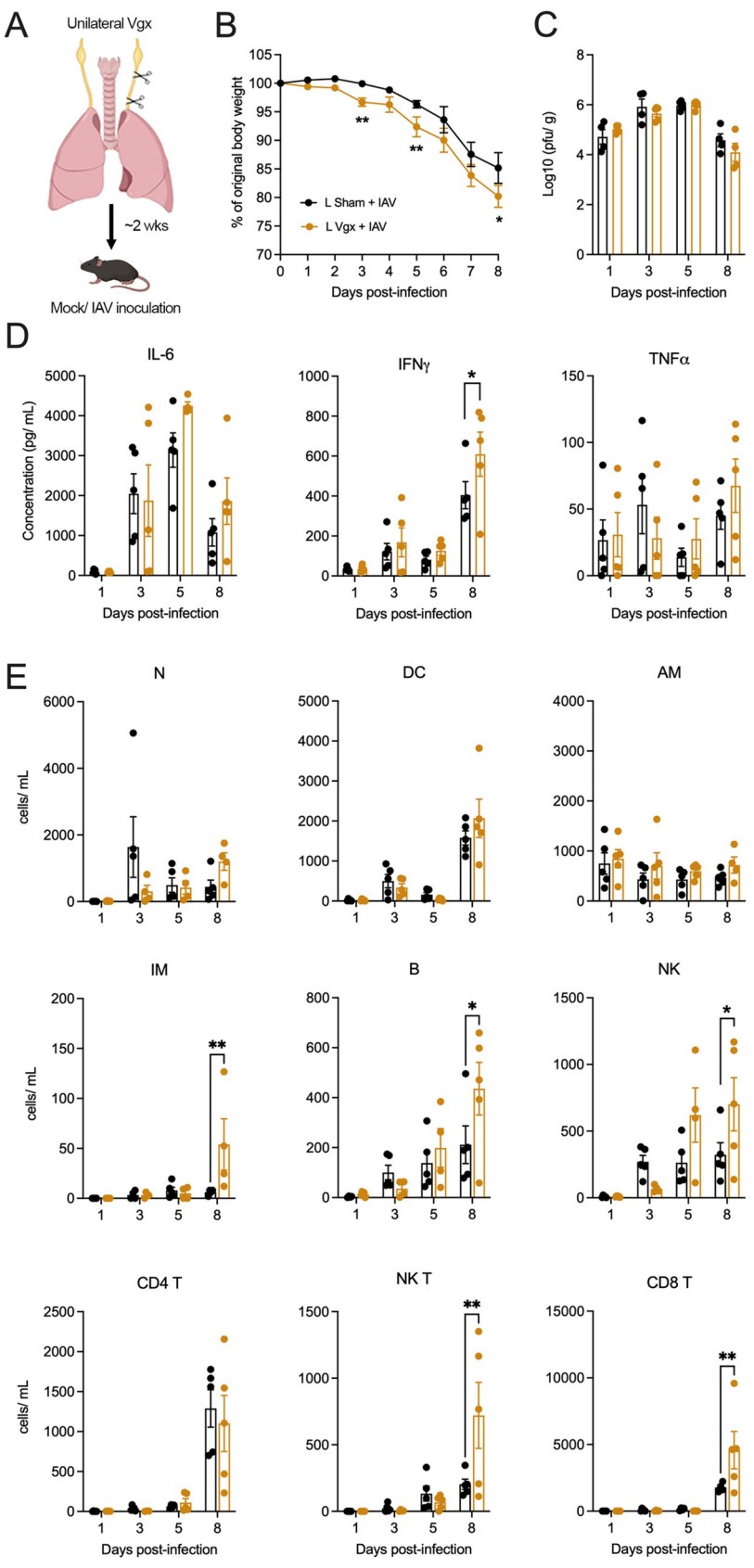

Fig 3. **The impact of left vagotomy on IAV induced pathogenesis.** (A) Schematic outlining the procedure for surgically removing partial vagal innervation to the lungs (left unilateral vagotomy). Graphs depict group level (B) body weight change, (C) lung viral titers, (D) lung cytokine measurements and (E) lung immune cell populations over the course of IAV infection following either right vagotomy (Vgx) or sham surgery (n = 5 per group at days 1, 3, 5, 8 post IAV infection). Data represented as mean ± SEM. *, ** denotes significance of $p < 0.05$, $p < 0.01$, respectively, as determined by repeated measures two-way ANOVA corrected for multiple comparisons (Šídák). N, neutrophils; DC, dendritic cells; AM, alveolar macrophages; IM, interstitial macrophages; B, B cells; NK, natural killer cells. Cartoons were created with BioRender.com.

over the course of the experiment between both mock infected groups (Fig 5D). In contrast, IAV infected mice regardless of treatment group, exhibited a decrease in body weight and increase in clinical signs, with the IAV infected QX-314 group displaying more significant changes compared to the IAV infected vehicle group (Fig 5B and 5C). IAV infection caused

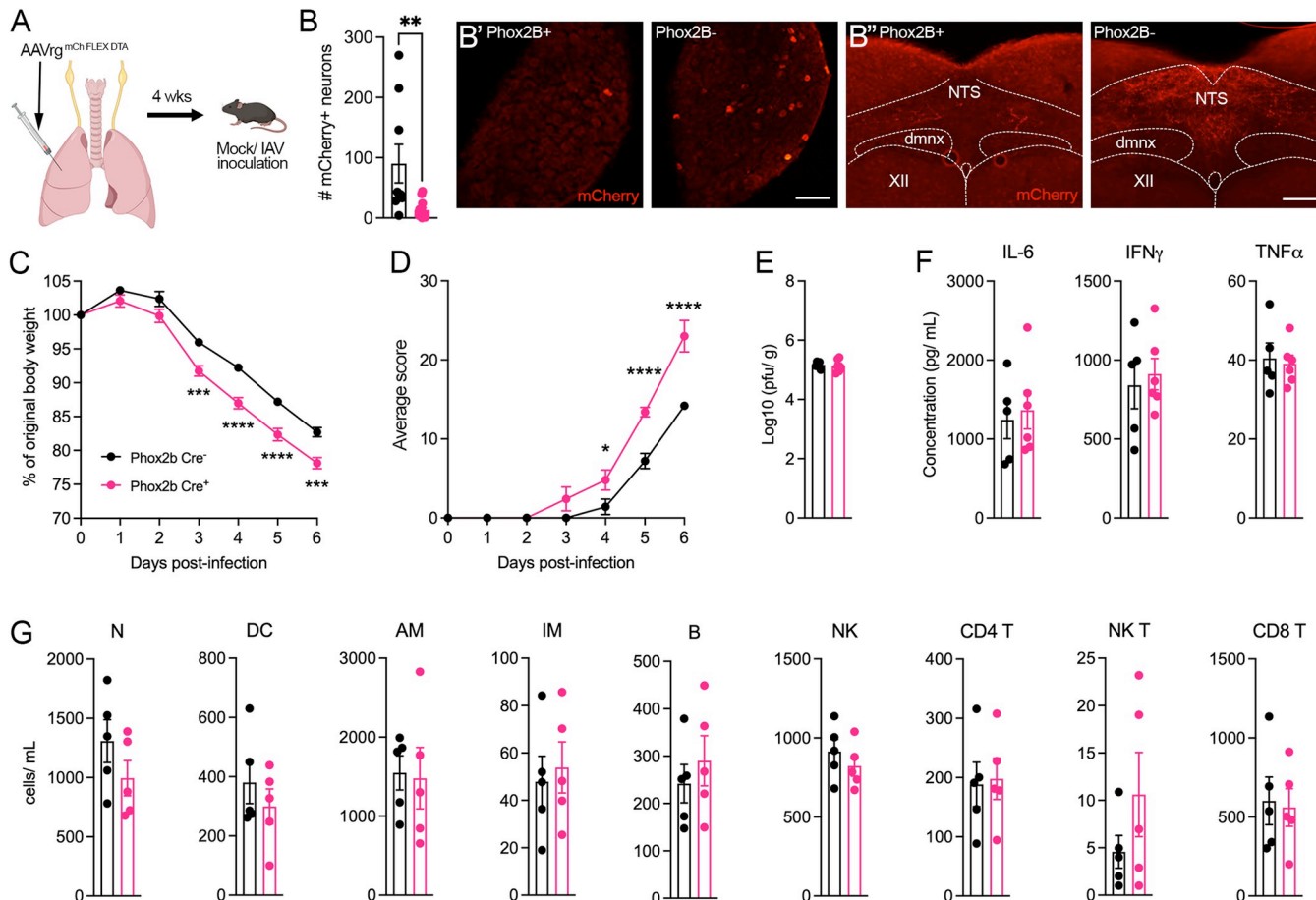

Fig 4. **The impact of genetic ablation of pulmonary vagal sensory neurons on IAV pathogenesis.** (A) Schematic showing genetic ablation of nodose lung sensory neurons in the vagal sensory ganglia using retrograde AAV encoding Cre-inducible diphtheria toxin (AAVrg mCherry-FLEX-DTA) in Phox2B-Cre+ (n = 6) and Phox2B-Cre- (n = 5) mice. (B) Quantification of the total number of mCherry+ neurons in the vagal sensory ganglia with (B' and B") showing representative images of the vagal sensory ganglia and brainstem nucleus of the solitary tract (NTS) taken from Phox2B-Cre+ and Phox2B-Cre- mice. Graphs depict group level (C) body weight change, (D) clinical score over the course of IAV infection, (E) lung viral titers, (F) lung cytokine measurements and (G) lung immune cell populations at day 6 IAV post infection following nodose lung sensory neuron ablation. Data represented as mean ± SEM. *, **, ***, **** denotes significance of $p < 0.05$, $p < 0.01$, $p < 0.001$, $p < 0.0001$, respectively, as (B, E, F, G) determined by Mann-Whitney t-test and (C, D) repeated measures two-way ANOVA corrected for multiple comparisons (Šídák). dmnx, dorsal motor nucleus of the vagus; XII, hypoglossal motor nucleus. N, neutrophils; DC, dendritic cells; AM, alveolar macrophages; IM, interstitial macrophages; B, B cells; NK, natural killer cells. Scale bar represents 100 μm. Cartoons were created with BioRender.com.

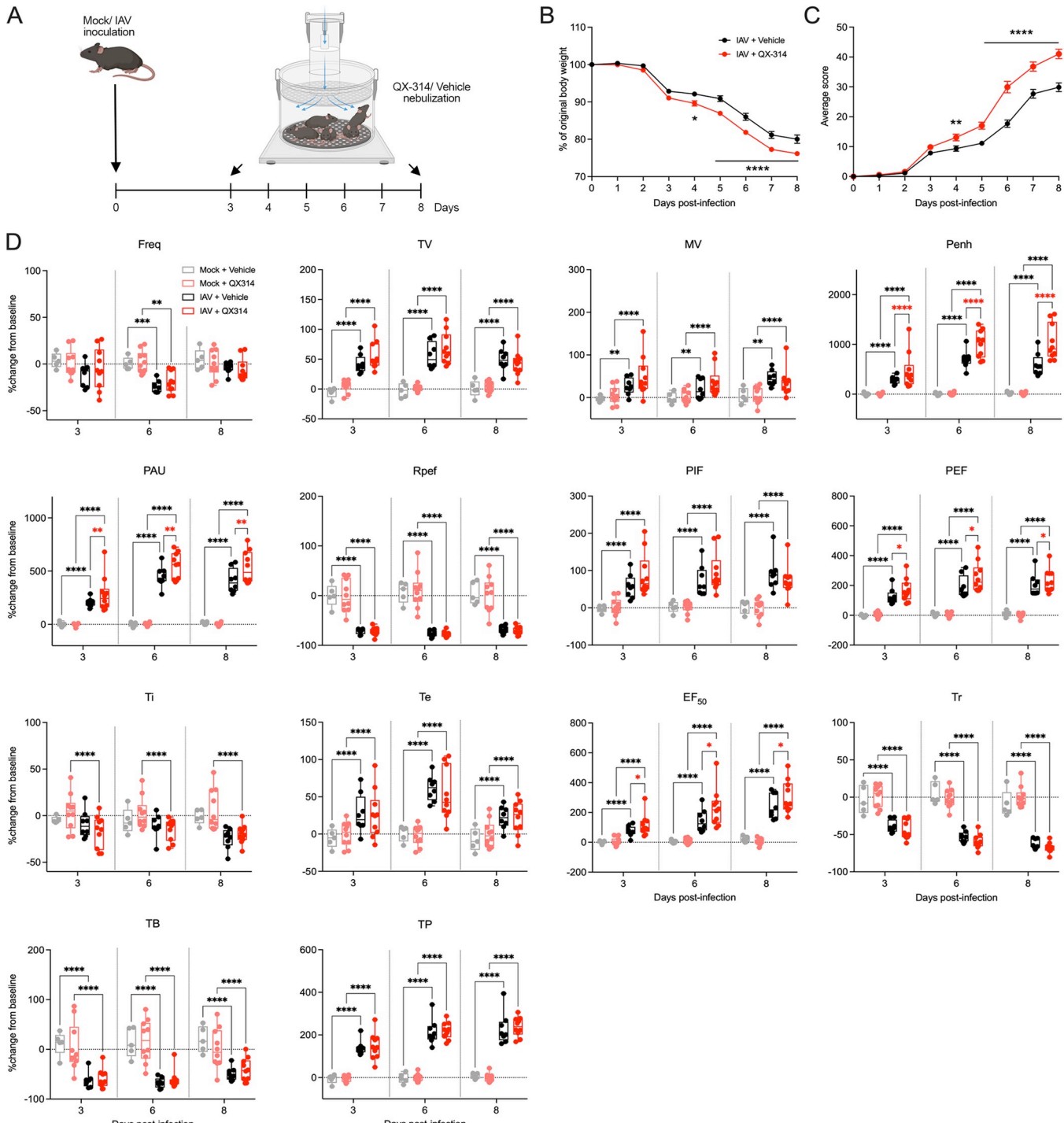

**Fig 5. Pharmacological inhibition of pulmonary sensory neuron activity during IAV infection increases morbidity.** (A) Schematic outlining experimental design used to assess the effect of nebulized QX-314 on IAV morbidity. Graphs depict group level (B) body weight change, (C) clinical scoring and (D) respiratory parameters measured post infection with IAV following administration of QX-314 or vehicle. N = 10 for each group at 4, 6, 8 days post infection. Data represented as mean ± SEM. *, **, ***, **** denotes significance of $p < 0.05$, $p < 0.01$, $p < 0.001$, $p < 0.0001$, respectively, as (B) determined by repeated measures two-way ANOVA corrected for multiple comparisons (Šidák). Freq, frequency; TV, tidal volume; MV, minute volume; PENH, enhanced pause; PAU, pause; Rpef, location into expiration where the peak occurs (PEF) as a fraction of Te; PIF, peak inspiratory flow; PEF, peak expiratory flow; Ti, inspiratory time; Te, expiratory time; EF50, expiratory flow at 50% expired volume; Tr, relaxation time; TB, duration of breaking; TP, duration of pause before expiration. Cartoons were created with BioRender.com.

significant changes in respiratory function in both vehicle and QX-314 mice compared to mock infected mice. As disease progressed, we observed significant decreases in respiratory rate (Freq), ratio of time to peak expiratory flow (Rpef), relaxation time (Tr) and duration of breaking (TB). Significant increases in tidal (TV) and minute volume (MV), Pause (PAU) and enhanced pause (Penh), peak inspiratory and expiratory flow (PIF and PEF), expiratory time (Te), ejection fraction 50 (EF50), and duration of pause before inspiration (TP) were also observed (Fig 5D), overall, suggestive of a change in respiratory function similar to acute respiratory distress associated with viral infection in humans. In IAV infected QX-314 mice, we observed a significant decrease in inspiratory time (Ti) compared to mock infected QX-314 mice and interestingly, IAV infected QX-314 mice showed significant increases in EF50, Pause, Penh and PEF (Fig 5D) compared to IAV infected vehicle mice. EF50 is the flow rate at which 50% of tidal volume has been expelled and has been shown to increase along with disease severity [38,39]. Penh is a non-specific measurement of breathing and is calculated based on = (PEF/ PIF) x pause. Therefore, the increases observed in PEF, Pause and PIF remaining unchanged likely lead to the increases seen in Penh. Again, this measurement has been shown to increase along with disease severity [38,40,41].

No differences were observed in proinflammatory lung cytokines, lung histopathology (S8C and S8D Fig) or immune cells present in the BAL (S8E Fig) between mock infected QX-314 and vehicle mice. In IAV infected mice, across both QX-314 and vehicle groups, there was an obvious increase in proinflammatory cytokines in the lungs (IL-6, IL-1β, IFNα,γ,λ and TNFα; Fig 6A), and lung viral load (Fig 6B), accompanied by changes in lung immune cell populations (Fig 6C), and significant lung pathology (Fig 7). When comparing QX-314 and vehicle IAV infected mice, we observed a significant decrease in IL-1β and IFNγ (Fig 6A), a reduction in the number of interstitial macrophages and CD8 T cells (Fig 6C), plus a small improvement in overall lung histopathology (Fig 7C) all at day 8 post-infection. These data suggest that QX-314 administration leads to an altered regulation of the IAV immune response in the lungs. However, the nature of this pulmonary neuroimmune regulation appears inconsistent with the worsening of clinical signs observed when vagal sensory neurons are disrupted, or their activity prevented.

## Pharmacological inhibition of pulmonary sensory neurons impacts IAV neuropathology

We have previously shown that severe pulmonary IAV infections in mice are accompanied by a neuroinflammatory phenotype in the vagal sensory ganglia [19,20]. To investigate whether peripheral sensory neuron activation in the lungs is involved in the development of vagal neuroinflammation, we compared the expression of inflammatory genes and inflammatory cell densities in the vagal sensory ganglia of IAV infected mice administered via inhalation from day 3 post IAV infection with either vehicle or QX-314 (Fig 5A). Consistent with our previous studies, in vehicle administered mice, IAV compared to mock infection resulted in the increase of expression of several inflammatory related genes within the vagal sensory ganglia, *Tmem173* (fold change: day 8, 2.21 ± 0.26, $p < 0.05$), *Il1b* (fold change: day 8, 6.27 ± 1.52, $p < 0.05$), *Irf9* (fold change: day 4, 2.07 ± 0.17, $p < 0.01$; day 8, 2.44 ± 0.14, $p < 0.0001$), *and Isg15* (fold change: day 8, 12.26 ± 3.43, $p < 0.01$). Inflammatory gene changes were accompanied by a significant increase in the number of ganglia MHC II+ immune cells of IAV compared to mock infected mice at day 6 and day 8 post-infection (IAV + vehicle: day 6, 58.8 ± 2.1% and day 8, 73.6 ± 2.4%; Mock + vehicle: 38.9 ± 1.9% MHC II+ cells relative to total neurons; $p = 0.0011$ (D6) and $p < 0.0001$ (D8)).

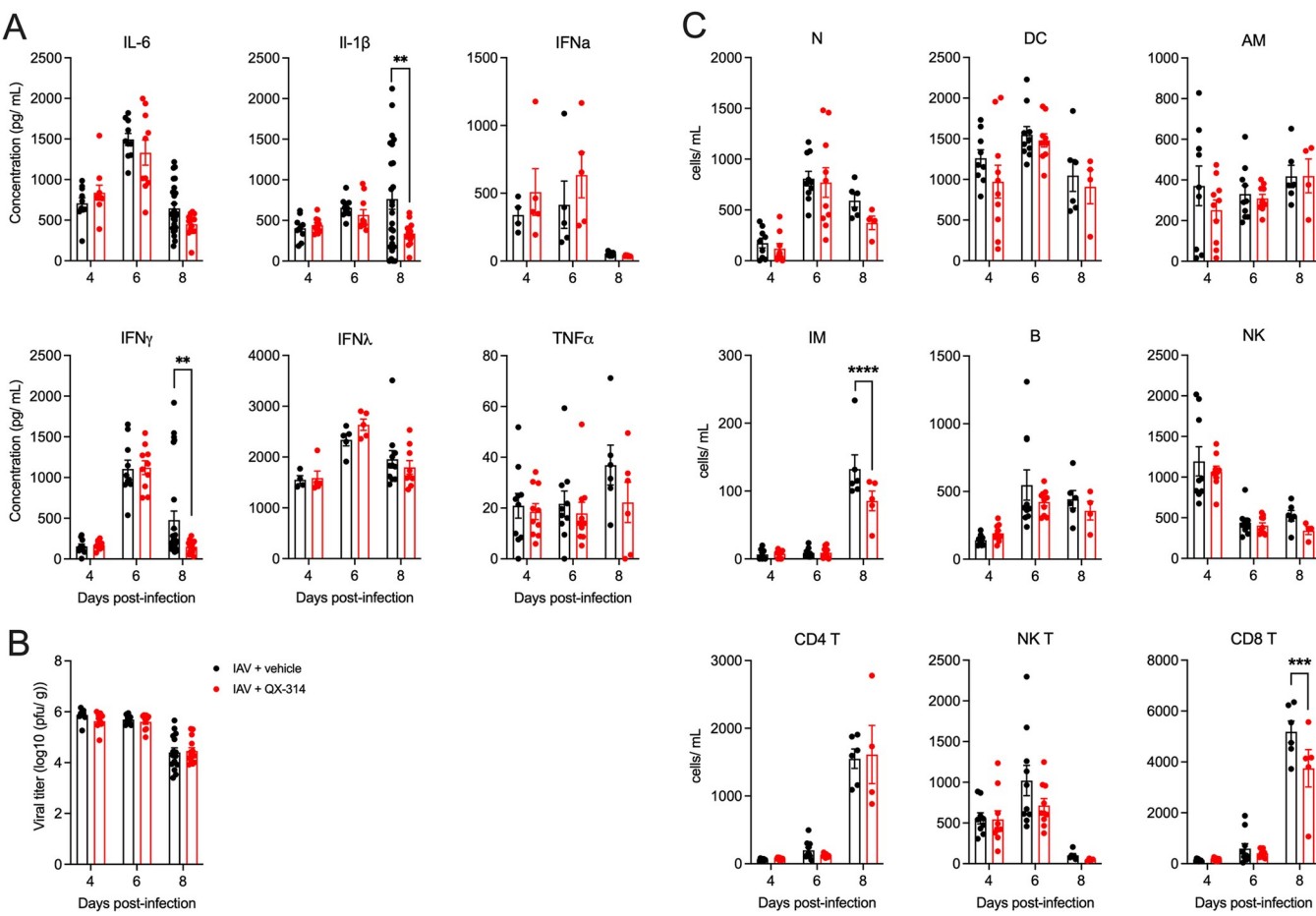

**Fig 6. Pharmacological inhibition of pulmonary sensory neuron activity impacts IAV lung pathogenesis.** Graphs depict group level (A) lung cytokines, (B) lung viral titers post infection and (C) lung immune cell populations IAV following administration of QX-314 or vehicle. N = 5–10 for each group at 4, 6, 8 days post infection. Data represented as mean ± SEM. *, **, ***, **** denotes significance of $p < 0.05$, $p < 0.01$, $p < 0.001$, $p < 0.0001$, respectively, as determined by repeated measures two-way ANOVA corrected for multiple comparisons (Šídák). N, neutrophils; DC, dendritic cells; AM, alveolar macrophages; IM, interstitial macrophages; B, B cells; NK, natural killer cells.

QX-314 administration resulted in an altered neuropeptide and inflammatory gene profile in the vagal sensory ganglia. In IAV infected QX-314 administered mice, we observed a significant increase in the expression of genes encoding the neuropeptides CGRP (*Calca*) and Substance P (*Tac1*) at both days 4 and 8 post infection compared to IAV vehicle mice. We also observed a further upregulation of genes associated with host defense against pathogens, *Tmem173* and *Il1b*, and a decrease in genes associated with an interferon antiviral response, *Irf9* and *Isg15* (Fig 8A). The H1N1 subtype of IAV is typically considered non-neurotropic, yet as we have previously reported [19] we detected viral mRNA in the vagal sensory ganglia at days 4 and 6 post infection of IAV infected mice administered vehicle. Intriguingly, mice administered QX-314 demonstrated lower ganglia viral mRNA levels, suggestive of lower infection rates or improved viral clearance (Fig 8B), perhaps consistent with the hyperinflammatory ganglionic environment. Mock infected mice administered either vehicle or QX-314 showed no difference in the number of MHC II+ immune cells and CGRP-expressing neurons within their vagal sensory ganglia (S8A and S8B Fig). However, IAV infection resulted in a further increase in the number of MHC II+ cells with mice that were administered QX-314

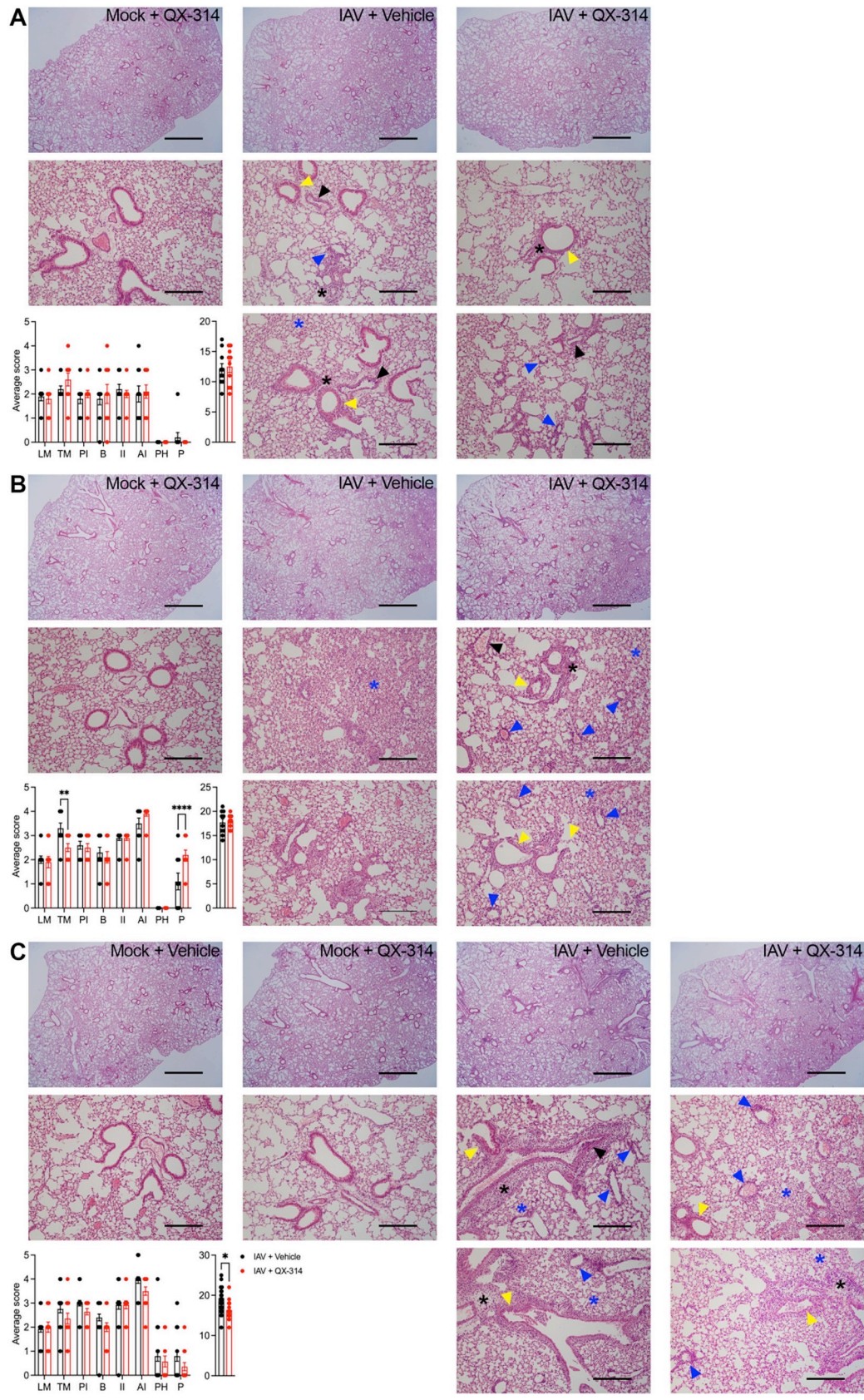

**Fig 7. Pharmacological inhibition of pulmonary sensory neuron activity during IAV infection and impacted lung pathology.** Images and represent hematoxylin and eosin-stained lung sections obtained from mice administered either QX-314 or vehicle at days (A) 4, (B) 6 or (C) 8 post mock or IAV infection. Graphs represent histopathological lung scoring of each criteria (left) and the subsequent total score (right) for QX-314 or vehicle administered IAV infected mice at days (A) 4, (B) 6, or (C) 8 post IAV infection. Data represented as mean ± SEM. *, **, **** denotes significance of $p < 0.05$, $p < 0.01$, $p < 0.0001$, respectively, as determined by repeated measures two-way ANOVA corrected for multiple comparisons (Šídák). Interstitial leukocyte infiltration (black asterisk), alveolitis (blue asterisk), leukocyte and transmural margination (black arrowheads), bronchiolitis (yellow arrowhead), perivascular lymphocyte infiltration (blue arrowheads). LM, leukocyte margination; TM, transmural migration; PI, perivascular infiltration; B, bronchitis, bronchiolitis; II, interstitial inflammation; AI, alveolar inflammation; PH, pneumocyte hypertrophy/ hyperplasia; P, pleuritis. Scale bar at low magnification 1,500 μM and high magnification 300 μM.

displaying a significantly higher number of MHC II+ immune cells at days 6 and 8 post-infection compared to IAV infected vehicle mice (Fig 8C). We also observed a significant increase in the overall number of CGRP-expressing neurons within the vagal sensory ganglia at days 4 and 6 post-infection in QX-314 mice compared to their vehicle counterparts (Fig 8D).

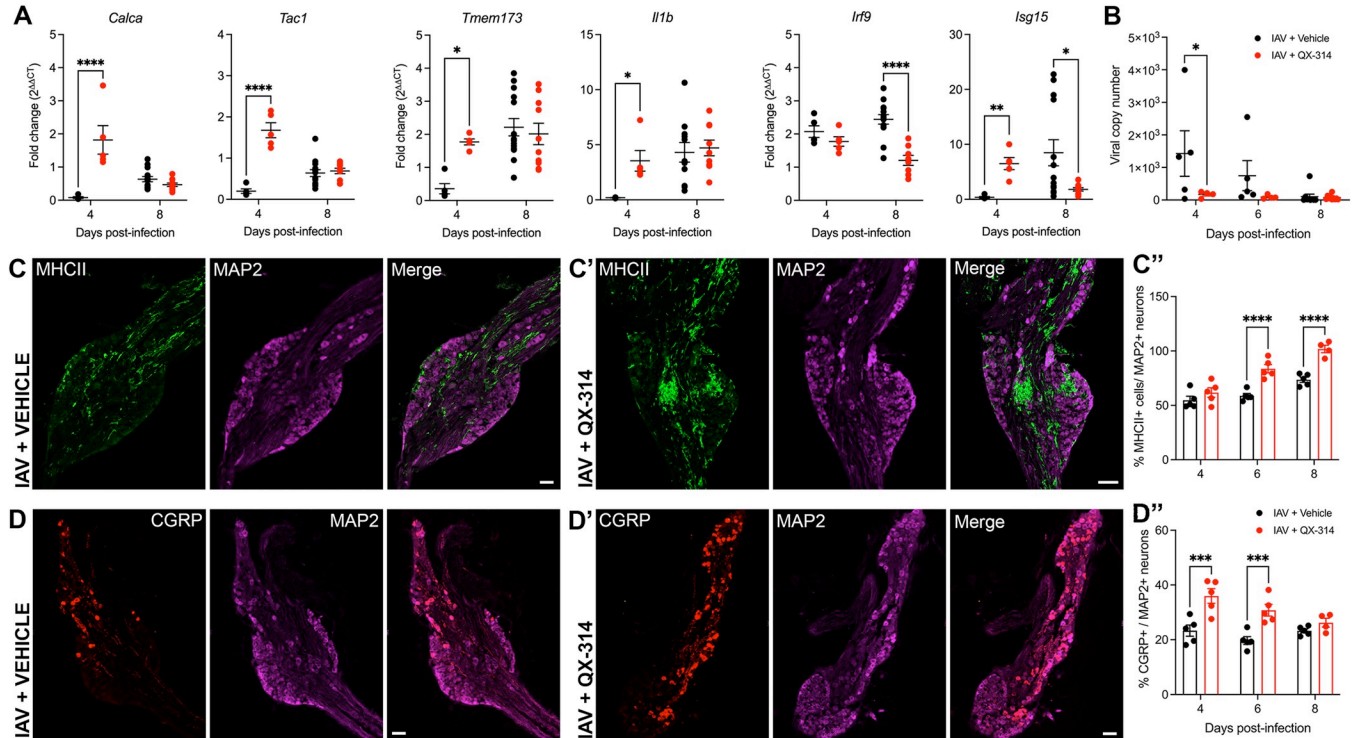

**Fig 8. Pharmacological inhibition of pulmonary sensory neuron activity during IAV infection results in a hyperinflammatory state within the vagal sensory ganglia.** Graphs depict group level qPCR analysis of (A) neuropeptide and host defense/ proinflammatory associated genes, and (B) viral mRNA levels present in the vagal sensory ganglia during IAV infection in mice that received either vehicle (black circles) or QX-314 (red circles). Data expressed as fold change in expression values relative to matched mock group. Representative immunofluorescence images of the vagal sensory ganglia at day 6 post-infection demonstrating an increase in the number of MHC II+ immune cells in mice administered either (C) vehicle or (C') QX-314 (MHC II, green; sensory neurons immunolabeled with MAP2, purple). The bar graph (C") shows the percentage increase of MHC II+ cells per total number of MAP2+ neurons in the vagal sensory ganglia of vehicle (black circles) and QX-314 (red circles) IAV infected mice. Representative immunofluorescence images of the vagal sensory ganglia demonstrating an increase in the number of CGRP-expressing neurons in IAV infected mice that received (D') QX-314 compared to mice receiving vehicle (D). The bar graph (D") shows the percentage increase of CGRP-expressing neurons (red) per the total number of MAP2 neurons (purple) in the in the vagal sensory ganglia of vehicle (black circles) and QX-314 (red circles) IAV infected mice. Data represented as mean ± SEM. *, **, ***, **** denotes significance of $p < 0.05$, $p < 0.01$, $p < 0.001$, $p < 0.0001$, respectively, as (A, B, C", D") determined by repeated measures two-way ANOVA or mixed-effects analysis corrected for multiple comparisons (Šídák). Scale bar represents 100μm.

## Discussion

There is a growing recognition that neuroimmune crosstalk contributes to the pathogenesis and clinical presentation in many diseases, potentially representing a novel therapeutic checkpoint for improved disease management. The present study advances this proposition by demonstrating that vagal neural processes contribute to the regulation of inflammation and morbidity during IAV infection. Denervation of the vagal sensory neural innervation to the lung prior to IAV infection resulted in worsened clinical appearance and marginally increased some IAV lung measures of pathology. Pharmacologically preventing pulmonary sensory neuron activation during IAV infection similarly increased clinical measures of disease severity, however, this was accompanied by a marginal improvement of lung measures of inflammation and pathology. IAV infection was accompanied by the development of a neuroinflammatory profile within the vagal sensory ganglia, which was worsened by inhibition of pulmonary sensory neuron function during IAV infection. Collectively, these data build on our understanding of neuroimmune interactions and suggest complex roles for the nervous system in pulmonary viral disease.

Vagal pathways have been shown to regulate pulmonary inflammation in dichotomous ways. For example, in a murine model of allergic asthma the inflammatory cytokine IL-5 was shown to trigger the release of the neuropeptide vasoactive intestinal protein (VIP) from a subset of pulmonary sensory neurons which, in turn, perpetuated inflammation by acting on CD4[+] T cells and innate lymphoid cells, increasing the production of TH2 cytokines associated with driving asthmatic conditions. Ablating these pulmonary sensory neurons reduced asthma severity [21]. The same authors showed that blocking the activity of pulmonary sensory neurons and the subsequent release of substance P during allergic asthma was sufficient to reduce allergy-induced goblet cell hyperplasia and hypersecretion of Muc5AC [22], a mucin that contributes to airway hyperactivity [42]. CGRP released from vagal lung sensory neurons during pulmonary staphylococcal infection has been shown to mediate immunosuppression, reducing neutrophil recruitment and lowering clearance of *S. aureus*. CGRP administration increased the bacterial load in the lungs of infected mice and worsened the clinical signs of pneumonia while sensory neuron ablation improved bacterial clearance and survival [23]. These observations suggest an important regulatory of pulmonary sensory neurons pulmonary inflammatory responses, which is perhaps specific to the underlying pathogen or disease pathology.

To understand the regulatory role of pulmonary sensory neurons in inflammatory processes during IAV infection, it was important to understand the innervation of the pulmonary system by the vagal ganglia. Our study shows the bilateral innervation of the lung lobes by the left and right vagal ganglia, albeit asymmetrical as the left lung lobe is predominantly innervated by the left vagal sensory ganglia, whereas the right lung lobes receive equal innervation by both ganglia. We and others have previously described asymmetry in vagal innervation and physiology, suggesting the right and left vagal nerves contribute differently to the processing of interoceptive signals [25,43–46]. Consequently, variation in contribution of the left and right vagus nerves to inflammatory regulation during IAV infection may exist. Consistent with this, in our study, although left and right unilateral vagotomy during IAV infection led to marginal increases in lung inflammatory pathology, the specific measures impacted by vagotomy differed. In accordance with our findings, Gao and colleagues also reported that vagotomy increased lung inflammation during IAV infection, seemingly related to reduced lung anti-inflammatory responses during a PR8 infection in mice. This effect was presumably due to a disruption of vagal contribution to the systemic anti-inflammatory reflex which is governed by autonomic innervation to splenic and other abdominal immune tissues [47]. Similar effects of vagotomy have also been reported during non-pathogenic systemic lipopolysaccharide

exposure [48]. By contrast, vagotomy has been shown to reduce pulmonary inflammation associated with lung fibrosis [49]. These findings suggest that vagal mechanisms may contribute to inflammatory regulation via both local tissue and systemic pathways, perhaps complicating interpretation of such studies. Indeed, vagotomy non-specifically disrupts all sensory and motor innervation to the tissues below that vagotomy level, and therefore we cannot presume that the effects observed in our vagotomy studies are specifically related to either sensory pathways or the innervation to the lungs. However, it is notable that systemic cytokines released during IAV infection were not altered by vagotomy, which may argue against a substantive disruption of the systemic anti-inflammatory reflex. Furthermore, anti-inflammatory effects were observed following QX-314 inhalation treatment in mice exposed to IAV, which presumably preserves the efferent pathways regulating the systemic anti-inflammatory reflex.

The genetic and pharmacological approaches that we employed to specifically ablate or inhibit pulmonary sensory neurons provided additional evidence for vagal neural control over IAV pathogenesis. A component of this effect may involve changes in sensory-nerve dependent regulation of pulmonary inflammation given that sensory neuron silencing (with QX-314) marginally improved measures of lung pathology. We and others have previously observed an increase in neuropeptide gene and protein expression markers in vagal sensory neurons following pulmonary viral infections [19] and administration of QX-314 in IAV infected animals elicited a further increase in vagal ganglia neuropeptide content in the present study. Regardless of the mechanism, it is intriguing that the anti-inflammatory effects of vagal sensory nerve manipulations in the present study did not impact viral load in the lungs. Thus, the change in lung inflammation following sensory nerve ablation or silencing is unlikely to be the only mechanism contributing to altered IAV pathogenesis. IAV infected animals consistently presented with greater weight loss and worsened clinical scores following vagotomy, sensory neuron ablation or inhibition, seemingly incongruent with the moderately altered pulmonary inflammatory effects and unchanged infectious burden.

In our previous study, we characterized transcriptomic changes and an increase in inflammatory cells in the vagal sensory ganglia following pulmonary infection with IAV, consistent with IAV-induced neuroinflammation [19]. Upregulated genes were mostly related to pathogen host defense and antiviral responses, typically downstream of interferon signalling, suggesting that pulmonary sensory neurons are influenced by the local antiviral inflammatory environment in the lungs during IAV infection. Here, we replicated these findings, and extend our prior observations by providing further evidence that vagal ganglia neuroinflammation is likely regulated through mechanisms that involve sensory neuron activation in the lungs. Thus, the expression of the interferon-related signalling genes *Irf9* and *Isg15* in the vagal sensory ganglia was reduced in animals receiving QX-314, conceivably related to the concomitant reduction in lung IFNγ, known to activate the heterotrimer Stat1:Stat2:Irf9 complex (interferon stimulating gene factor 3; ISGF3) to enhance transcription of interferon stimulated genes (ISGs) and pro-inflammatory cytokines [50,51].

Intriguingly, however, the effects of QX-314 on vagal ganglia neuroinflammation were not as predicted, as inflammatory cell numbers in the vagal ganglia were significantly increased in IAV-infected animals receiving QX-314. In addition, several other pro-inflammatory genes were upregulated in IAV infected animals administered QX-314, including the inflammasome associated genes *Il1b* and *Tmem173* (encodes for stimulator of interferon genes; STING) [52,53]. The inflammasome in neurons is likely a defense mechanism against viral replication [54,55] and whilst H1N1 IAV is not typically considered a neurotropic virus, previous studies have reported the presence of viral RNA in neural tissue such as the vagal ganglia and pulmonary sensory neurons following pulmonary infection [19]. Perhaps QX-314 administration may lead to enhanced activation of sensory neuron inflammasome signalling to combat IAV

reaching and replicating within the vagal ganglia. Indeed, the viral copy number of IAV was decreased in the vagal sensory ganglia of animals that received QX-314. The potentiation of the increased number of ganglia immune cells (MHC II expressing) and upregulated neuropeptide expression could also be part of an increased immune response against IAV neuronal invasion. However, we cannot rule out these findings reflect an alternative response to, or cause of, the transcriptomic changes through currently unknown mechanisms. Regardless, the altered neuroinflammatory profile following inhibition of the activity of pulmonary sensory neurons with QX-314 may result in broad neurological impacts and underpin the altered clinical presentation evident in these animals.

Although this study provides evidence for a role of vagal sensory processes in the pathogenesis of IAV infections, several considerations about the approaches we employed warrant discussion. The interventions in the present study were not designed to fully deprive the lungs of regulation by sensory innervation. Unilateral vagotomy preserves some bilateral lung innervation, as shown by our retrograde neuronal tracing data. Similarly, AAV DTA-mediated ablation of bronchopulmonary sensory neurons is unlikely to be 100 percent efficient as it relies upon uptake and retrograde transport of AAV following intraluminal injections in the lung, which need to balance the delivery of sufficient virus to the airways to reach deeper lung regions while avoiding virus reaching off-target extrapulmonary tissues. The level of QX-314 anaesthesia will also vary in relation to drug pharmacodynamics and pharmacokinetics and given our clinically-relevant twice daily treatment regime beginning after IAV infection was established, we cannot rule out an important role of sensory neurons in infection establishment at earlier time points. Accordingly, our data may under-represent the role of vagal sensory systems in regulation IAV disease. Additionally, we cannot exclude a role for other cell types in the intervention responses observed in the present study. This is notable for vagotomy where vagal preganglionic motor neurons may also contribute to observed effects, but also for QX-314 where off-target effects due to possible (gastrointestinal) ingestion of the nebulized drug or actions on non-neuronal cells in the airways have not been widely explored.

In conclusion, these data show that the pulmonary vagal sensory neurons play a role in the regulation of immune responses in the pulmonary system during IAV infection. Surgical, genetic, and pharmacological interventions targeting the vagus nerve and sensory neurons had modest effects on lung inflammation and viral pathogenesis but worsened the clinical presentation perhaps due to a dysregulated vagal neuroinflammation. The exact mechanisms by which this occurs remains to be elucidated. Whether the neuroinflammatory impacts of IAV pathogenesis extend beyond the vagus nerve into the central nervous system and circuits regulated by pulmonary vagal sensory neurons warrants further investigation. Conceivably, central mechanisms could contribute to the myriad of disease behaviours, such as malaise, myalgia, and cough that accompany IAV and other severe lung infections [56,57]. Targeting specific vagal sensory pathways could thus be a potential therapeutic approach to improve IAV disease outcomes.

## Materials and methods

### Ethics statement

This study complies with the *Australian Code for the Care and Use of Animals for Scientific Purposes* from the National Health and Medical Research Council of Australia. All procedures were approved by the Animal Ethics Committee of both The University of Queensland, Australia and the University of Melbourne, Australia (approval number 071/17 and 20986, respectively).

## Virus details

Viral stocks of Influenza Auckland/ 1/ 2009 (Auck/ 09; H1N1) and Puerto Rico/ 8/ 34 (PR8, H1N1) were propagated as previously described in embryonated chicken eggs, approved by the University of Queensland Animal Ethics Committee (AE000089). Viral titres of IAV were determined by plaque assay on Madin-Darby Canine Kidney (MDCK) cells, as previously described [58]. We note the use of the PR8 strain of H1N1 with the vagotomy experiments and the Auck/ 09 H1N1 strain with the experiments involving pharmacological inhibition of neural activity. An initial characterization revealed no differences of IAV pathogenesis between the two viral strains (S10 Fig) and we also observed a comparable neuroinflammatory profile within the vagal sensory ganglia (Fig 8 and [19]).

## Murine model

**Mouse strains.** All steps were taken to minimize the animals' pain and suffering, as mandated in the Australian Code. Mice were housed in temperature controlled (21°C), individually ventilated cages on a 12-hour light/ dark cycle, with access to pelletized food and water ad libitum in groups up to five per cage. Pathogen-free wildtype C57BL/6JArc mice (8–10 weeks of age, male) were obtained from the Animal Resources Centre (ARC; Western Australia, Australia) and used for all experiments involving surgical denervation, retrograde neuronal tracing and QX-314 treatment. Heterozygous Phox2b-Cre mice (Stock No: 016223; B6(Cg)-Tg (Phox2b-Cre)3Jke/J) were initially purchased from the Jackson Laboratory, with a colony bred and maintained at the University of Melbourne in accredited facilities. Phox2b mice expressing Cre (Phox2b-Cre+) and littermate controls (Phox2b-Cre-) used for experimentation (8–12 weeks of age, male and female) were generated by crossing male heterozygous Cre mice to female wildtype C57BL/ 6 mice. Mice were genotyped for Cre as per genotyping guidelines on the Jackson Laboratory website.

**Surgical procedures.** A unilateral vagotomy was performed to partially disrupt nerve supply to the lungs. Mice were anesthetized with gaseous isoflurane (4% induction, 1.25–1.5% maintenance) and once their withdrawal and palpebral reflexes had disappeared a midline incision was made on the ventral surface of their neck to expose the right or left mid-cervical vagus nerve. Care was taken to carefully dissect each vagus nerve from the carotid artery and cervical sympathetic nerve. To avoid reinnervation approximately 7mm segment of nerve was removed, the wound was sutured, and mice were allowed to recover for at 10–14 days during which time body weight was monitored and post-operative analgesia was administered for 48 hours (Meloxicam, 5mg/ kg and buprenorphine 0.05mg/ kg s.c). A control sham group receiving the surgical procedure (everything described above omitting the removal of the vagus nerve) were also included. All mice, on average had reached pre-surgical body weight on the day of IAV infection (S1B Fig).

To determine the laterality of vagal sensory innervation to the right and left lung lobes we performed a series of experiments utilizing the retrograde viral tracer, AAV CAG-tdTomato (AAVrg$^{TdT}$; Addgene 59462-AAVrg; 3μl of 4.6 x $10^{12}$ GC/ ml). Mice were split into 3 groups whereby group 1 had both vagi intact, group 2 had left vagus nerve intact (right vagotomy), and group 3 had right vagus nerve intact (left vagotomy). Within each group mice were further split into 2 subgroups whereby they received an injection into either the left or right lung lobes. Briefly, mice were injected from the posterior side and through the seventh intercostal space as previously described [15]. Skin and external intercostal musculature was retracted, the lung visualized underneath the intact internal intercostals and, using a 30 G needle attached to a 10 μl gastight Hamilton syringe, 3 μl of AAVrg$^{TdT}$ virus was injected into the lung lobe. The needle was left in place for 2 min to limit the spread of AAV onto surrounding tissues. The

incisions were sutured, and animals were allowed to recover for 21 days during which body weight was monitored daily and post-operative analgesics administered (see above).

Intraganglionic microinjection of 500 nl AAVretro CAG-mCherry-FLEX-DTA (AAVrg$^{mCh-FLEX-DTA}$; 4.85 x 10$^{11}$ vg/ ml) into the vagal sensory ganglia of Phox2B-Cre mice was performed. Briefly, mice were anesthetized with isoflurane (4% induction, 1.25%-1.5% maintenance) and placed in a supine position on a thermostatically controlled surgical mat set at 36.5˚C. To access the vagal sensory ganglia, a midline incision in the skin was made to expose the submandibular gland, then the sternocleidomastoid and omohyoid muscles were retracted to expose the carotid sheath. The cervical vagus nerve was gently detached from the carotid artery and sympathetic nerve until the caudal portion of the vagal sensory ganglia was visible (around where the superior laryngeal nerve exits from the vagus nerve). The hypoglossal nerve was carefully detached from the vagus nerve and retracted rostrally to gain a clear view of the vagal sensory ganglia. Microinjections into the right vagal sensory ganglia were performed via glass pulled micropipettes (tip diameter ~20 μM) mounted onto a micromanipulator and connected to a microprocessor controlled picopump (model PV820, World Precision Instruments, USA). Following completion of microinjections, the skin wound was closed with sterile suture. Mice were allowed to recover for a minimum 28 days to allow sufficient cellular ablation and recovery from surgery, during which body weight was monitored and post-operative analgesia was administered for 48 hours (see above).

To genetically ablate lung nodose sensory neurons, Phox2B-Cre mice received an injection of 6 μl AAVrg$^{mCh-FLEX-DTA}$ diluted in 10 μl of sterile saline, slowly (over 2 minutes) into the tracheal lumen ~5–6 cartilage rings below the larynx via a 10 μl Hamilton syringe connected to a 30 G needle, the needle was left in place for 30 s to prevent backflow following which the incision was sutured. Animals were allowed to recover for 28 days during which body weight was monitored daily and post-operative analgesics administered (see above).

**IAV infection.** Mice were infected intranasally (50 μl) with 50 PFU A/Puerto Rico/8/1934 H1N1 (PR8) or 5.5 x10$^3$ PFU of A/Auckland/1/2009 H1N1 (Auck/09) suspended in phosphate buffered saline (PBS, pH 7.4, Thermo Fisher, Australia) under isoflurane-induced anaesthesia (1–3 L/ min) with PBS being used for mock infections. Over the course of disease, mice were monitored at the same time every day for body weight changes and clinical signs associated with disease progression. The scoring criteria included in the clinical signs assessment include:

1. Percentage weight loss calculated from pre-infection weight (no change OR 1–20% loss of body weight OR > 20% loss of body weight).

2. General condition, which included appearance of fur (shiny and unruffled OR slightly ruffled OR ruffled and matte), eye condition (clear, clean, and open OR unclean, evidence of discharge, semi-closed/ closed), posture (normal OR slightly hunched OR moderately hunched OR severely hunched), and clinical complications (paralysis, tremor AND/ OR vocalizations, breathing sounds AND/ OR animal is appearing cold to touch, decrease in body temperature).

3. Motility (spontaneous normal behavior with social contacts OR spontaneous but reduced OR moderately reduced behavior OR motility only after stimulation OR isolation, coordination disorders, not responding to stimulation).

4. Respiration (breathing normal OR breathing slightly changed i.e., hyper/ hypoventilation, labored/ shallow breathing/ gasping, up to 10% change OR breathing moderately altered, 10–30% change OR breathing strongly altered, 30–50% change).

**QX-314 treatment.** Mice were nebulized with lidocaine N-ethyl bromide (N-(2,6-Dimethylphenylcarbamoylmethyl) triethylammonium bromide (QX-314; Sigma Aldrich, Australia) dissolved in sterile saline (300μM) or vehicle (sterile saline), twice daily (12 hrs apart) beginning at three days post IAV infection. This consisted of mice (groups of five) placed in a Buxco small animal whole body plethysmography chamber (diameter, 23cm; height, 12 cm; volume, 4985 cm$^3$) and exposed to aerosolized QX-314 or vehicle (Buxco Aerosol Distribution Unit 5 LPM, Aerogen nebulizer unit AG-AL1000 with filter cap of 3.1 μm particle size; Data Sciences International) over 20 minutes, at an air flow velocity of 2.5 L/ min and at 50% nebulizer duty cycle. The inhaled dose of QX-314 for each animal was, on average 0.04 mg/ kg. This value was calculated using this algorithm, in accordance with [59]: Inhaled dose (mg/kg) = [C (mg/ L) x RMV (L/ min) x D (min)]/ BW (kg), where C is the concentration of drug in air inhaled, RMV is respiratory minute volume, D is the of exposure in minutes, BW is bodyweight in kg, with RMV (in L/ min) calculated as 0.608 x BW (kg)$^{0.852}$.

**Whole-body plethysmography recordings.** Respiratory parameters were measured in mice (QX-314 experimental group) using calibrated 4-chamber unrestrained whole-body plethysmography (WBP, Buxco, Wilmington, USA). Mice were acclimatized to the plethysmography chambers (30 minutes per day), each day for 3 days prior to experimentation. Respiratory frequency (f, breaths min $^{-1}$), tidal volume (TV, ml kg $^{-1}$), minute volume (MV, ml min $^{-1}$), inspiratory and expiratory time (Ti and Te, sec), estimated peak inspiratory and expiratory flow (PIF and PEF, ml sec $^{-1}$), pause and enhanced pause (PAU and Penh, arbitrary unit of measure), the location into expiration where the peak occurs (PEF) as a fraction of Te (Rpef), expiratory flow at 50% expired volume (EF50, ml sec -1), relaxation time (Tr, sec), duration of breaking–the percentage of breath occupied by transitioning from inspiration to expiration (TB), and duration of pause before expiration–percentage of breath occupied by transition from expiration to inspiration (TP) were automatically sampled from the calibrated Buxco flow trace, corrected for animal body weight and ambient chamber temperature, and calculated in real time by the plethysmography software (Buxco Finepointe Software Version 2.1.0.9). Real time data collected every 2 sec were used to calculate the average per 20 min for each animal at baseline (day 0, prior to infection) and days 4, 6, 8 post-infection. Data was normalized to baseline values for each animal. Normalization was performed to remove confounding factors such as inter-animal variation in baseline parameters.

**Tissue sampling and analyses.** All tissue samples were obtained from mice that were euthanized post-infection with sodium pentobarbital (100 mg/ ml) and exsanguinated unless otherwise stated.

**Viral plaque assay.** Lung lobes were homogenized in DMEM (ThermoFisher Scientific, Australia), clarified by centrifugation, and stored at -80˚C until analysis. Viral titers in clarified tissue homogenate samples were determined by plaque assay, as described previously [60].

**Cytokine measurements.** Cytokine levels in clarified tissue lung homogenates were determined by enzyme linked immunosorbent assay (ELISA) according to the manufacturer's instructions (BD Biosciences, USA) and read with CLARIOstar Plus (BMG labtech) microplate reader. Serum cytokine levels were determined using the Anti-virus Response panel of LEGENDplex Multiplex Assays (BD Biosciences, USA) according to the manufacturer's instructions, measured using the Cytoflex S flow cytometer (Beckman Coulter, USA) and analysed using LEGENDplex Qognit software (BD Bioscience, USA).

**Broncho-alveolar lavage fluid (BALF) immune cell populations.** BALF samples were clarified by centrifugation with red blood cells (RBC) lysed in RBC lysis buffer (0.15 M $NH_4Cl$, 17 nM Tris-HCl at pH 7.2, sterile filtered) for 1 min and cell pellet resuspended in 0.1M PBS. Samples were treated with BD Fc Block (Rat Anti-mouse CD16/CD32; BD Biosciences, USA) for 20 min at 4˚C, followed by incubation in the following fluorophore conjugated anti-mouse

antibody combinations for 20 min at 4˚C: FITC NK1.1 (clone PK136), PerCP-Cy5.5 anti-CD8a (clone 53–6.7), PE/ Cy7 anti-CD3 (clone 53–6.7) (BD Biosciences, USA); FITC anti-CD11b (clone M1/ 70), PE anti-CD4 (clone GK1.5), PerCP-Cy5.5 anti-GR-1 (clone RB6-8C5), APC anti-F4/ 80 (clone BM8), APC anti-B220 (clone RA3-6B2) (Biolegend, USA); PE/Cy7 anti-CD11c (clone N418) (eBioscience, USA) and eFluor780 fixable viability dye (Thermo-Fisher Scientific, Australia). This was followed by fixation using BD Cytofix Fixation buffer (BD Biosciences, USA) to preserve immunofluorescent staining until analysed. Samples were analysed using the Cytoflex LX flow cytometer (Beckman Coulter, USA) and data analysed with FlowJo_v10 software. Immune cell populations were classified as followed: alveolar macrophages (F4/80$^+$ CD11b$^{int/low}$ CD11c$^{high}$), interstitial macrophage (F4/80$^+$ CD11b$^{high}$ CD11c$^{int}$), neutrophils (F4/80$^-$ GR-1$^{high}$ CD11b$^{high}$), dendritic cells (F4/80$^-$ GR-1$^-$ CD11b$^{high}$ CD11c$^{high}$), CD4$^+$ T cells (CD3$^+$ CD4$^+$), CD8$^+$ T cells (CD3$^+$ CD8$^+$), NK cells (CD3$^-$ NK1.1$^+$), NKT cells (CD3$^+$ CD4$^-$ CD8$^-$ NK1.1$^+$). See S11 Fig for gating strategy.

**Pulmonary histology.** Lung lobes were collected and fixed in 10% formalin for a minimum of 24 h then transferred to 70% ethanol for processing by the Core Histology Facility, Translational Research Institute (Woolloongabba, Queensland, Australia). Lung samples were embedded in paraffin wax prior to sectioning at 7 μm using a Hyrax M25 Rotary Microtome (Leica Biosystem, Germany). Sections were stained with hematoxylin and ethanol-based eosin and assessed by a veterinary pathologist who was blinded to the study designs. Each sample was scored for vascular changes, bronchitis/ bronchiolitis, interstitial inflammation, alveolar inflammation, pneumocyte hypertrophy and pleuritis in a semi-quantitative manner on a scale of 0–5 where 0 = no change, 1 = minimal change, 2 = mild change, 3 = moderate change, 4 = severe change in < 50% of lung lobe and 5 = severe change in > 50% of lung lobe. See S12 Fig for examples of scoring criteria.

**Vagal sensory ganglia RNA isolation and quantitative PCR.** Vagal sensory ganglia analyses were performed on mice as part of the QX-314 experiments (see above). Mice were euthanized at day 4, 8 post-infection and bilateral vagal sensory ganglia freshly harvested, placed in 250 μl TRIzol reagent (Qiagen, Australia), snap frozen and stored at -80˚C until further processing. Thawed samples were homogenized and RNA was extracted and concentrated using the RNeasy MiniElute Cleanup kit (Qiagen, Australia) according to the manufacturer's instructions. cDNA was generated using the high-capacity cDNA reverse transcription kit (Life technologies) and random primers (host RNA) according to the manufacturer's instructions. Real-time PCR was performed on generated cDNA with SYBR Green using QuantStudio 6 Real-Time PCR (ThermoFisher Scientific, Australia). Forward and reverse prime sequences for each gene of interest are shown in Table 1. Gene expression was normalized relative to Hypoxanthine phosphoribosyl transferase (*Hprt*) and β-actin (*Actb*) expression. Fold

**Table 1. Primers used for vagal sensory ganglia qPCR.**

| GOI | Forward primer | Reverse primer |
|---|---|---|
| *Actb* | GACTCCTATGTGGGTGACGAGG | GGATCTTCATGAGGTAGTCCGTCA |
| *Hprt* | AGGCCAGACTTTGTTGGATTTGAA | CAACTTGCGCTCATCTTAGGCTTT |
| *Tac1* | AGAGCTTTAAATTCTGTGGC | GAATAGATAGTGCGTTACAGG |
| *Calca* | TTGAGGTCAATCTTGGAAAG | CTTTCATCTGCATATAGTCCTG |
| *Isg15* | ATGGAGGACTTTTGGGATAG | AGAGGCAGAGCTTTTTATTG |
| *Irf9* | CTACTTCTGTAGAGATTTGGC | GATGAGATTCTCTTGGCTATG |
| *Il1b* | GGATGATGATGATAACCTGC | CATGGAGAATATCACTTGTTGG |
| *Tmem173* | CTCATTGTCTACCAAGAACC | TAACCTCCTCCTTTTCTTCC |

change was calculated based on the $2^{\Delta\Delta CT}$ method. Viral copy number was determined using the A/Puerto Rico/8/1934 H1N1 virus matrix gene cloned into the pHW2000 plasmid as previously described [61]. Primers used for IAV quantification, forward primer: AAGACCAATCT TGTCACCTCTGA, reverse primer: TCCTCGCTCACTGGGCA.

**Immunofluorescence assay of vagal sensory ganglia.** A subset of mice from the QX-314 experiments and Phox2B-Cre mice were euthanized with (100 mg/ kg, i.p.) sodium pentobarbital and transcardially perfused with PBS; 0.1 M, pH 7.45, followed by 4% PFA; pH 7.45. Vagal sensory ganglia and brainstems were removed and post-fixed in 4% PFA (2 hours) then cryoprotected in 30% sucrose (w/v PBS) overnight before freezing in Optimal Cutting Temperature compound (OCT, TissueTek). Twelve μm sections were collected on a cryostat (-20°C), sequentially over 4 slides (Superfrost Plus) for each pair of vagal ganglia from one animal, dried at room temperature for 2 hrs and stored at -80°C. 50 μm cryostat cut sections of Phox2B-Cre brainstems encompassing the medullary region were collected serially into 0.1M PBS and processed for immunohistochemistry that same day. All sections were rinsed in PBS then blocked in 10% donkey serum in 0.1 M PBS for 1 hr, before incubation for 24 hours in the primary antibody of interest: Rat anti I-A/ I-E, 1:1000, Biolegend, 107601; Chicken anti-microtubule associated protein 2 (MAP2), 1:1000, Abcam, Ab5392; Rabbit anti-calcitonin gene-related peptide (CGRP), 1:3000, Immunostar, 24112; Rabbit anti- dsRed, 1:1000, Clontech, 632496; Rabbit anti- neurofilament 200kD (NF 200kD), 1:1000, Sigma Aldrich, N4142; Sheep anti-tyrosine hydroxylase (TH), 1:1000, Merck Millipore, AB1542; Rabbit anti- purinergic receptor 2 (P2X2), 1:500, Alomone, APR-003; Goat anti- choline acetyl transferase, 1:200, Chemicon, AB144P. All antibodies were diluted in antibody diluent (2% donkey serum and 0.3% Triton X-100 in PBS). After the required incubation, sections were washed three times with PBS for 20 mins each then incubated for one hour in appropriate fluorescently-conjugated secondary antibodies (1:500, ThermoFisher Australia). Brainstem sections were kept in order and mounted onto gelatin-coated slides. All slides were coverslipped with an antifade mounting media (ProLong Gold, ThermoFisher Scientific).

Sections were visualized on a fluorescent microscope (Leica DM6 B) with high resolution images taken with Leica DFC7000 T camera (x100) at the same exposure for each protein of interest across each sample for offline analysis. For vagal sensory ganglia, counts of inflammatory cells, neurons with MAP2 and CGRP, NF 200kD and TH or mCherry positive neurons were performed offline on all sections. Brainstem sections from Phox2B-Cre mice were viewed for the presence of either P2X2 positive or mCherry positive fibers within the NTS or CHAT positive neurons within the dmnx. Neuron counts performed within the dmnx spanned 800μM (bregma -8.00 to -7.20mm). High resolution digital images taken at 200x magnification were imported into Adobe Photoshop (version 20.0.4) and optimized (minimally) for brightness, contrast, and size for preparation of representative photomicrographs.

## Statistical analysis

Data are expressed as the mean ± SEM unless stated differently. Statistical analyses were performed using GraphPad Prism, version 8.3.1 for Windows (GraphPad Software, CA, USA). Normality of test groups were determined using Shapiro-Wilk's test. Statistical significance for normally distributed data was determined using a two-tailed, unpaired students t-test (between two groups). A Mann-Whitney test was used for data that was not normally distributed. If one or more repeatedly measured variables were considered, uni- or multi-factorial repeated-measures ANOVA was performed. Statistical significance was set at $p \leq 0.05$. See text or figure legends for more details.

## Supporting information

**S1 Fig. Evidence for lack of reinnervation after cervical vagotomy and body weight measurements in mice following unilateral vagotomy or sham surgery.** Graphs depict (A) quantification of the total number of tdTomato+ neurons (retrogradely labelled from either left or right lung lobes) in either left or right vagal sensory ganglia (VG) following unilateral left or right vagotomy and (B) body weight change post vagotomy or sham surgery and prior to IAV or mock infection. Data represented as mean ± SEM.
(TIFF)

**S2 Fig. Body weight, lung cytokine measurements and lung immune cell populations in vagotomized and sham mice following mock infection.** Graphs depict body weight change, lung cytokine measurements and lung immune cell populations in (A, C, E) right and (B, D, F) left vagotomized/ sham mice following intranasal mock (PBS) inoculation. N = 5 for each group at 8 days post infection. Data represented as mean ± SEM.
(TIFF)

**S3 Fig. Serum cytokine measurements in vagotomized and sham mice following IAV infection.** Graphs depict cytokine measurements taken from vagotomized and sham mice following IAV or mock infection (day 5). N = 5 for each group. Data represented as mean ± SEM.
(TIFF)

**S4 Fig. Confirmation of AAV encoding Cre-inducible diphtheria toxin (AAVrg$^{\text{mCh FLEX DTA}}$) fucntionality.** (A) Schematic showing intraganglionic injection of genetic ablation of sensory neurons in the vagal sensory ganglia AAVrg$^{\text{mCh FLEX DTA}}$ in Phox2B-Cre$^+$ (n = 7) and Phox2B-Cre$^-$ (n = 3) mice. (B) Graphs depict the percentage of (A) neurofilament (NF) 200kD and (B) tyrosine hydroxylase (TH) -expressing neurons per the total number of neurons (MAP2+) in either right (R; injected side) or left (L; non injected side) vagal ganglia (VG) and images (C) demonstrate immunostained vagal ganglia sections from the right vagal ganglia from either Phox2B-Cre$^+$ or Phox2B-Cre$^-$ mice following 4 weeks of microinjection of AAVrg$^{\text{mCh FLEX DTA}}$. Data represented as mean ± SEM. *, ***, **** denotes significance of $p < 0.05$, $p < 0.001$, $p < 0.0001$, respectively, as (B) determined by mixed-effects analysis corrected for multiple comparisons (Šídák). Scale bar represents 100μm. Cartoons were created with BioRender.com.
(TIF)

**S5 Fig. Characterization of AAV encoding Cre-inducible diphtheria toxin (AAVrg$^{\text{mCh FLEX DTA}}$) in the medullary brainstem.** Images represent coronal sections of medullary brainstem obtained from individual Phox2B-Cre$^+$ (M1-5) and Phox2B-Cre$^-$ (M6-8) mice microinjected with AAVrg$^{\text{mCh FLEX DTA}}$ into the right vagal sensory ganglia. Brainstem sections are immunostained for the purinergic receptor 2 (P2X2) to label vagal sensory nerve terminals within the nucleus of the solitary tract (NTS) and choline acetyl transferase (CHAT) to label preganglionic neuron soma within the dorsal motor nucleus of the vagus (dmnx). L, denotes left side and R, denotes right side (injected). AP, area postrema; cc, central canal; XII, hypoglossal motor nucleus. Scale bar represents 100 μm. Numbers in merge images represent bregma level.
(TIF)

**S6 Fig. Body weight and respiratory measurements following cellular ablation of lung pulmonary vagal afferents with AAVrg$^{\text{mCh FLEX DTA}}$.** Graphs depict (A) body weight change post-surgery after injection of AAVrg$^{\text{mCh FLEX DTA}}$ into the lungs of Phox2B-Cre$^+$ or Phox2B-Cre$^-$ mice and (B) respiratory parameters measured 30 days post-surgery (prior to

IAV infection; n = 5 per group). Data represented as mean ± SEM. Freq, frequency; TV, tidal volume; MV, minute volume; PENH, enhanced pause; PAU, pause; Rpef, location into expiration where the peak occurs (PEF) as a fraction of Te; PIF, peak inspiratory flow; PEF, peak expiratory flow; Ti, inspiratory time; Te, expiratory time; EF50, expiratory flow at 50% expired volume;Tr, relaxation time; TB, duration of breaking; TP, duration of pause before expiration. (TIF)

**S7 Fig. Body weight and lung immune cell population measurements following cellular ablation of lung pulmonary vagal afferents with AAVrg$^{mCh\ FLEX\ DTA}$ in mock-infected Phox2B-Cre$^+$ and Phox2B-Cre$^-$ mice.** Graphs depict (A) body weight change and (B) lung immune cell populations (Day 6 post-infection) in Phox2B-Cre$^+$ and Phox2B-Cre$^-$ mice following intranasal (PBS) inoculation. N = 3–5 per group. Data represented as mean ± SEM. (TIF)

**S8 Fig. Body weight, clinical score, lung pathology, lung cytokine and lung immune cell population measurements in QX-314 and vehicle treated mice following mock infection.** Graphs depict (A) body weight change, (B) clinic scoring, (C) lung histopathology, (D) lung cytokine and € lung immune cell population measurements in mice receiving nebulized 300 μM QX-314 or vehicle (saline) following intranasal mock (PBS) inoculation. N = 5–10 for each group at days 4, 6, 8 post infection. Data represented as mean ± SEM. N, neutrophils; DC, dendritic cells; AM, alveolar macrophages; IM, interstitial macrophages; B, B cells; NK, natural killer cells. (TIFF)

**S9 Fig. Immunohistochemical analysis of MHC II expressing immune cells and CGRP expressing neurons within the vagal sensory ganglia of QX-314 and vehicle treated mice following mock infection.** Graphs depict the percentage of (A) MHC II-expressing cells and (B) CGRP-expressing neurons per the total number of neurons in the vagal sensory ganglia in mice receiving nebulized 300 μM QX-314 or vehicle (saline) following intranasal mock (PBS) inoculation. N = 5 for each group at days 4, 6, 8 post infection. Data represented as mean ± SEM. (TIFF)

**S10 Fig. Comparison of IAV H1N1 viral strains.** Graphs depict (A) lung cytokines, (B) lung viral titers, and (C) lung histopathological score post infection with either Auck/ 09 or PR8 H1N1 viral strains. N = 5–7 for each group at 3, 6, 8 days post infection. Data represented as mean ± SEM. (TIF)

**S11 Fig. Flow cytometry gating strategy.** Representative flow cytometry plots used to identify immune cell populations in mouse bronchoalveolar lavage fluid (BALF). (TIF)

**S12 Fig. Lung histopathology scoring criteria for murine influenza A virus induced pneumonia.** Representative images of hematoxylin and eosin stained sections of lung tissue from mice infected with IAV demonstrating (A, A') leukocyte margination (black arrowheads), transmural margination (black arrows) and perivascular infiltration (blue arrows). (B) Bronchiolitis consisting of transepithelial leukocyte infiltration, accumulation of cellular debris in bronchiolar (Br) lumen (blue arrowhead). (C) Interstitial inflammation (yellow asterisk). (D, D') Alveolar inflammation (alveolitis) consisting of infiltration of neutrophils (blue asterisk), monocytes, macrophages and lymphocytes into alveolar spaces and (to a lesser degree) in the alveolar septae. (E) Pneumocyte hyperplasia consisting of type II pneumocytes becoming

enlarged and multiplying (mitosis, yellow arrowhead). (F) Pleuritis consisting of hypertrophy of mesothelial lining (green arrowheads) and subpleural infiltration of neutrophils, monocytes and fewer lymphocytes. Alveolitis present in the underlying parenchyma (black asterisk). Abbreviations: Ar, arteriole and Al, alveoli. Black scale bar represents 150 μm.
(TIF)

## Acknowledgments

The pHW2000 plasmid used for quantifying IAV infection in tissue samples was sourced and distributed kindly from St. Jude Children's Research Hospital. Cartoons were created with BioRender.com.

## Author Contributions

**Conceptualization:** Nathalie A. J. Verzele, Stuart B. Mazzone, Alice E. McGovern.

**Data curation:** Nathalie A. J. Verzele, Brendon Y. Chua, Helle Bielefeldt-Ohmann, Stuart B. Mazzone, Alice E. McGovern.

**Formal analysis:** Nathalie A. J. Verzele, Brendon Y. Chua, Alice E. McGovern.

**Funding acquisition:** Brendon Y. Chua, Stuart B. Mazzone, Alice E. McGovern.

**Investigation:** Nathalie A. J. Verzele, Brendon Y. Chua, Kirsty R. Short, Aung Aung Kywe Moe, Isaac N. Edwards, Helle Bielefeldt-Ohmann, Katina D. Hulme, Ellesandra C. Noye, Marcus Z. W. Tong, Matthew W. Trewella, Alice E. McGovern.

**Methodology:** Nathalie A. J. Verzele, Stuart B. Mazzone, Alice E. McGovern.

**Project administration:** Nathalie A. J. Verzele, Alice E. McGovern.

**Resources:** Kirsty R. Short, Patrick C. Reading, Stuart B. Mazzone, Alice E. McGovern.

**Supervision:** Kirsty R. Short, Stuart B. Mazzone, Alice E. McGovern.

**Validation:** Nathalie A. J. Verzele, Kirsty R. Short, Stuart B. Mazzone, Alice E. McGovern.

**Visualization:** Nathalie A. J. Verzele, Alice E. McGovern.

**Writing – original draft:** Nathalie A. J. Verzele, Stuart B. Mazzone, Alice E. McGovern.

**Writing – review & editing:** Nathalie A. J. Verzele, Brendon Y. Chua, Kirsty R. Short, Aung Aung Kywe Moe, Isaac N. Edwards, Helle Bielefeldt-Ohmann, Katina D. Hulme, Ellesandra C. Noye, Marcus Z. W. Tong, Patrick C. Reading, Matthew W. Trewella, Stuart B. Mazzone, Alice E. McGovern.

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
