## [Decision Letter · Decision Letter 0]

2 Nov 2023

Dear Dr McGovern,

Thank you very much for submitting your manuscript "Evidence for vagal sensory neural involvement in influenza pathogenesis and disease" for consideration at PLOS Pathogens. As with all papers reviewed by the journal, your manuscript was reviewed by members of the editorial board and by several independent reviewers. In light of the reviews (below this email), we would like to invite the resubmission of a significantly-revised version that takes into account the reviewers' comments.

We cannot make any decision about publication until we have seen the revised manuscript and your response to the reviewers' comments. Your revised manuscript is also likely to be sent to reviewers for further evaluation.

Sincerely,

Jacob S. Yount

Academic Editor

PLOS Pathogens

Meike Dittmann

Section Editor

PLOS Pathogens

Kasturi Haldar

Editor-in-Chief

PLOS Pathogens

orcid.org/0000-0001-5065-158X

Michael Malim

Editor-in-Chief

PLOS Pathogens

orcid.org/0000-0002-7699-2064

Reviewer's Responses to Questions

**Part I - Summary**

Reviewer #1: The manuscript by Verzele et al. on the role of vagal sensory innervation of the lung during influenza infection provides some unique insight into the role of neuroimmune circuits in the lung. In general, the manuscript is well written, and is obviously a substantial amount of work that is presented. General comments for the authors to improve the manuscript include a revision to integrate some of these data better. For example, the wonderful tracing studies presented in Figure 1 are not prominently mentioned in the discussion with any interpretation or indication as to what these mean for the current study and going forward. The data on the immune cells and cytokines is similarly presented in my view. I didn’t get a feeling for what the changes in how the authors believed the different populations contributed to any changes. Personally, I did not like the many tables in this version and would have preferred separate graphs for each cell population so that one can easily see changes over time. Other suggestions include the removal of the precise numbers in the actual text, it simply repeats what is in the figure and for me disrupts the flow and decreases readability. In the end, I’m not sure what is the proposed mechanism here.

Specific questions are listed as minor and major with either the associated figure number or text line number.

Reviewer #2: The manuscript under review uses a mouse model of Influenza A virus to investigate how ablation of the vagal nerve or ablation of lung sensory neurons or inhibition of pulmonary sensory neurons impacts on the course of influenza A disease and inflammatory and immune responses.

They find a complex picture where vagotomy and ablation of pulmonary vagal sensory neurons has almost no effect on pulmonary inflammation yet worsens clinical disease and inhibition of pulmonary sensory neurons decreases lung pathology and worsens clinical disease.

Overall, this is a well-designed study using a very interesting approach. The methods are described thoroughly and the paper is well-written.

While this is certainly an interesting study, I have concerns about the success rate of the methods to manipulate vagal innervation and about the interpretation of the results.

1) Overall the effects are rather marginal and one wonders if the used methods to manipulate vagal innervation are really working efficiently. For the complete vagal denervation, the authors say that this led to a transient reduction in body weight (data not shown). Why are these data not shown? In the world of paperless publication, there is no reason for not showing these data. This also applies to the data for the spread of the AAV. In humans, vagotomy (which in the past was done also in the non-gastro selective way) leads to continuous weight loss. Why should this be different in mice? Could it be that there is reinnervation, which would put the murine vagotomy paradigm under stress?

2) In Fig. 1, D and G, the vagatomized mice with Influenza A lose more weight than the sham-operated ones. Could this be due to the vagal denervation per se. To exclude this one would need a group of mice with vagatomy alone. The same is true for the mice shown in Fig. 1L where ablation of pulmonary vagal sensory neurons is achieved by genetic tools.

3) In the discussion, the authors correctly state that:” These observations suggest that pulmonary sensory neurons may play complex and multifaceted roles in respiratory disease, perhaps specific to the underlying pathogen or disease pathology.”I think all of us would agree with that. The question what does this paper add to unravel this complex and multifaceted role…..

**Part II – Major Issues: Key Experiments Required for Acceptance**

Reviewer #1: Table 1, How are there interstitial macrophages in the BALF?, The gating strategy presented in figs5 does not seem to take note that both alveolar and interstitial macrophages express CD11c, with alveolar macrophages being CD64+ (or f480+) CD11b-/low CD11c+ SiglecF+, with interstitials typically CD64+ CD11b+ CD11c+ SiglecF-. I also noticed that the plots provided and gating strategy appear to be showing the height (H) not area (A) for each fluorphore. This certainly is not standard and makes further interpretation difficult. I did not find how these gates were set, and if a FMO control was used on the BALF. In my view the population after the F4/80 gate is likely all alveolar macrophage, and that SiglecF should be used here to discern these. I agree the CD11b staining is quite high and am not sure if this is biologically real, a consequence of the height display, or simply a detector voltage issue. Please see PMC3824047 table 1, and associated figures for other widely used gating strategies. Is there a reason why only BALF was assessed and not whole lung?

Table 1. I’m not sure what population is the most important in the author's mind or the phenomenon being described. For the significant populations, it would help to include these in a figure and have the rest of the data in a table. I would like to be able to cross-compare more quickly and visually these different data sets across time.

Fig1 Please include H&E figures even in supplementary data so that readers and reviewers can see these data.

Fig1/Text starting at line 153: The strategy to ablate Phox2B+ lung innervation is an interesting approach. Can you describe

more about how you validate that less mCherry is really due to loss of these cells? Can you perform ISH or confocal for phox2B+ and count cells in the NTS? The concern is that the lack of a virally encoded signal may not reflect that it is working as you expect. Was there any change in mouse body weight after virally mediate ablation of neurons in the NTS or is this really restricted and without effect on any other neurons and functions within the NTS?

Fig 1. With the virally mediated neuronal depletion, are there any changes in the vagus that would occur due to infection with your viral vector? Can you describe what was done to mitigate this possibility, or is the virally transduced phox2B.Cre- mouse completely normal?

Please indicate the time post-infection in the figure for the phox2b.Cre LSL-DTA mice experiments, also I didn’t see this information in table 4.

Again, there are indications in the results at line 178 about reduced pathology without the actual H&E as well.

Line 184/fig 2 for the QX314 studies, is there evidence that TRPV1 is only expressed on neurons especially during viral infection? Is there any data on this already in the literature?

Line 195. Nebulization of the drug was clearly intended to allow for exposure in the airway, is there some possibility that nebulized drug was ingested during animal grooming and would this alter your conclusions? Can you estimate the dose (ug/Kg of body weight) that the animals received?

Tables 5. These are some of the most interesting data that could use additional explanation in the discussion and how these fit with the ideas of neuronal regulation of pathology.

In Fig S2 why is vehicle group for all days of infection not shown?

In Fig 3. Please put some group/figure keys on the graphs. In the text at line 227 in describing cytokines is this in the lung or in the BALF. Will these two compartments match? Is there a role for IFNa/b or lambda here? Again tables of flow cytometry, the questions about gating strategy need to be addressed.

Line 249: This portion of the manuscript seems to be the focus of the authors, at this point it seems almost like the immune cell data and other measures are not as well integrated with the description of the effects on the nodose presented beginning at line 249. I also appreciate that Isg15 is an interferon stimulated gene, however, is this really because of the measured IFNg or that IFNa/b/l, that was not measured, is driving this expression?

Discussion. I think the sentence at line 287 that summarizes the work, misses some nuance. Perhaps I missed it but, I do not see the histopathology so I don’t know if I agree that pathology was slightly increased in denervation. Some aspects of clinical presentation were not changed such as body weight. These statements in my view are therefore an overstatement or that the data doesn’t seem to be presented.

Reviewer #2: see above

**Part III – Minor Issues: Editorial and Data Presentation Modifications**

Reviewer #1: Minor issues:

Line 127 Is the lung viral titer from the whole lung, Would there be differences in left vs. right lung lobe titers in mice with right or left vagotomy. Is there a hyper-local response to a viral pathogen or challenge?

Line 144: There is no difference in the three cytokines measured would be a statement that better reflects the data, not that systemic cytokines are the same.

Line 146, 149 on line 146 there is an indication that the table values are x10^3, Is this correct, and confirm that this is correct for all the other tables?

Line 518 flow gating is fig s5 not 4 as indicated. In this section you should better indicate how the gates were set. Are these FMO on BALF etc.

Reviewer #2: see above

PLOS authors have the option to publish the peer review history of their article (what does this mean?). If published, this will include your full peer review and any attached files.

Reviewer #1: No

Reviewer #2: No
---

## [Editor Report · Decision Letter 1]

1 Apr 2024

Dear Dr McGovern,

We are pleased to inform you that your manuscript 'Evidence for vagal sensory neural involvement in influenza pathogenesis and disease' has been provisionally accepted for publication in PLOS Pathogens.

Best regards,

Jacob S. Yount

Academic Editor

PLOS Pathogens

Meike Dittmann

Section Editor

PLOS Pathogens

Michael Malim

Editor-in-Chief

PLOS Pathogens

orcid.org/0000-0002-7699-2064
---

## [Editor Report · Acceptance letter]

9 Apr 2024

Dear Dr McGovern,

We are delighted to inform you that your manuscript, "Evidence for vagal sensory neural involvement in influenza pathogenesis and disease," has been formally accepted for publication in PLOS Pathogens.

Best regards,

Michael Malim

Editor-in-Chief

PLOS Pathogens

orcid.org/0000-0002-7699-2064